



# Atmospheric drag effects on modelled LEO satellites during the July 2000 Bastille Day event in contrast to an interval of geomagnetically quiet conditions

Victor U. J. Nwankwo[1], William Denig[2], Sandip K. Chakrabarti[3], Muyiwa P. Ajakaiye[1],
Johnson Fatokun[1], Adeniyi W. Akanni[1], Jean-Pierre Raulin[4], Emilia Correia[4], and John E. Enoh[5]

[1]Anchor University, Lagos 100278, Nigeria
[2]St. Joseph College of Maine, Standish, ME 04084, U.S.A
[3]Indian Centre for Space Physics, Kolkata 700084, India
[4]Centro de Rádio Astronomia e Astrofísica Mackenzie, Universidade Presbiteriana Mackenzie, São Paulo, Brazil
[5]Interorbital systems, Mojave, CA 93502-0662, U.S.A.

**Correspondence:** Victor U. J. Nwankwo (vnwankwo@aul.edu.ng)

**Abstract.** In this work we simulated the effects of atmospheric drag on two model SmallSats in Low Earth Orbit (LEO) with different ballistic coefficients during 1-month intervals of solar-geomagnetic quiet and perturbed conditions. The goal of this effort was to quantify how solar-geomagnetic activity influences atmospheric drag and perturbs satellite orbits. Atmospheric drag compromises satellite operations due to increased ephemeris errors, attitude positional uncertainties and premature satel-

lite re-entry. During a 1-month interval of generally quiescent solar-geomagnetic activity (July 2006) the decay in altitude ($h$) was a modest 0.53 km (0.66 km) for the satellite with the smaller (larger) ballistic coefficient of $2.2 \times 10^{-3}$ m²/kg ($3.03 \times 10^{-3}$ m²/kg). The associated Orbital Decay Rates (ODRs) during this quiet interval ranged from 13 m/day to 23 m/day (from 16 m/day to 29 m/day). For the disturbed interval of July 2000 the significantly increased altitude loss and range of ODRs were 2.77 km (3.09 km) and 65 m/day to 120 m/day (78 m/day to 142 m/day), respectively. Within the two periods more detailed

analyses over 12-day intervals of extremely quiet and disturbed conditions revealed respective orbital decays of 0.16 km (0.20 km) and 1.14 km (1.27 km) for the satellite with the smaller (larger) ballistic coefficient. In essence, the model results show that there was a 6-7 fold increase in the deleterious impacts of satellite drag between the quiet and disturbed periods. We also estimated the enhanced atmospheric drag effect on the satellites' parameters caused by the July 2000 Bastille Day event (in contrast to the interval of geomagnetically quiet conditions). The additional percentage increase due to the Bastille Day

event to the monthly mean values of $h$ and ODR are 34.69% and 50.13% for Sat-A, and 36.45% and 68.95% for Sat-B. These simulations confirmed; (i) the dependence of atmospheric drag force on a satellite's ballistic coefficient, and (ii) that increased solar-geomagnetic activity substantially raises the degrading effect of satellite drag. In addition, the results indicate that the impact of short-duration geomagnetic transients can have a further deleterious effect on normal satellite operations. While none of these findings were particularly surprising or profound we suggest that a model of satellite drag when combined with

a high-fidelity atmospheric specification, as was done here, can lead to improved satellite ephemeris estimates.





## 1 Introduction

Atmospheric drag describes the force exerted on an object moving through the atmospheric medium. The orientation of the drag force is in the reverse direction of relative motion with the resulting effect of impeding the motion of the object. Spacecraft moving through the atmosphere experience the atmospheric drag force which expends energy at the expense of the orbital motion (Wertz and Larson, 1999; Chobotov, 2002; Nwankwo, 2016). Atmospheric drag is the largest force affecting the motion of satellites in Low Earth Orbit (LEO) especially at altitudes below 800 km (Nwankwo et al., 2015). Space weather enhances atmospheric drag on satellites in LEO and the resultant impact can be profound (Nwankwo, 2016). Extreme space weather can cause satellite orbits to unexpectedly degrade making it more difficult to maneuver spacecraft and to identify and track satellites and other space debris (Nwankwo et al. (2015); and references therein). Another detrimental impact of enhanced satellite drag is the unplanned loss of otherwise healthy spacecraft due to premature atmospheric re-entry. Under this scenario a satellite would gradually decay from orbit (losing altitude) and would re-enter the earth's lower atmosphere unless appropriate orbit-raising maneuvers were implemented. Examples of spacecraft that prematurely re-entered the atmosphere include Skylab (launched 14 May 1973, re-entered 11 July 1979) and the Russian Radar Ocean Reconnaissance Satellites (RORSATs), Kosmos-954 (launched 18 September 1977, re-entered 24 January 1978) and Kosmos-1402 (launched 30 August 1982, re-entered 07 February 1983) (Nwankwo, 2016).

The orbital lifetime of a LEO satellite is subjected to the integrated atmospheric drag force experienced by the satellite over time. Figure 1 depicts the orbital degradation of a hypothetical satellite in a nominally circular orbit which, in this case, degrades over time from a starting position "A" at an altitude of 400 km to a later position "B" at an altitude below 110 km. The drag or negative acceleration, $f_d$, (units of m/s$^2$) experienced by the satellite is given as $f_d = \frac{1}{2}\rho B v_s^2$ where $\rho$ (units of kg/m$^3$) is the altitude-dependent atmospheric density and $v_s$ (units of m/s) is the satellite speed (King-Hele, 1987). The satellite speed, $v_s$, is a consequence of the balance between an inward-directed (towards earth) gravitational force at the satellite altitude and the outward-directed orbital centripetal force. A simplified version of a satellite's ballistic coefficient, $B$ (units of m$^2$/kg), is given as $B = C_d A_s / m_s$ where $C_d$ is the unit-less atmospheric drag coefficient, $A_s$ (units of m$^2$) is the satellite's projected area in the direction of motion and $m_s$ (units of kg) is the satellite mass (Bowman, 2002; Bhatnagar et al., 2005). For altitudes representative of most LEO satellites, that being between 140 km to 400-600 km, a constant drag coefficient, $C_d$, of 2.2 is appropriate (Cooke, 1965). A drag force, $F$ (units of kg m/s$^2$), acting in opposition to the satellite's motion, is given as $F = -m_s f_d$. In this work we model changes in the Orbital Decay Rate (ODR, units of m/day) and the monthly mean orbital decay (units of km) experienced by 2 satellites having different ballistic coefficients, $B$, under different solar-geomagnetic conditions (Nwankwo et al., 2020).





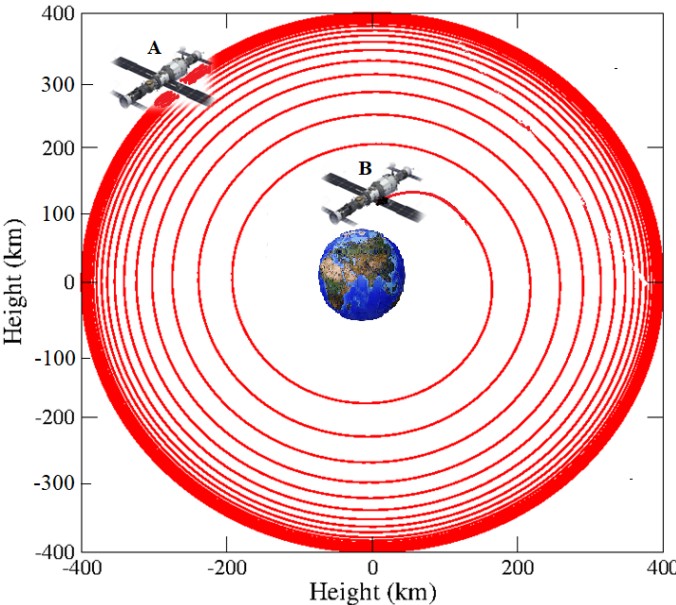

**Figure 1.** Orbital decay scenario of a LEO satellite due to atmospheric drag. The satellite orbit decreases from altitude "A" to altitude "B" as a consequence of atmospheric drag (Adapted from Nwankwo (2016)).

## 2 Solar and geomagnetic activities and their implications for atmospheric drag

Solar activity describes changes in the overall energy and mass output from the sun consisting of both long-term trends within the 11-year solar cycle (longer term changes are beyond the scope of this effort) and transient events of increased solar output.
Electromagnetic radiation (light) is continuously emitted from the sun across a broad spectral range from energetic gamma rays to radio-waves (Eddy, 2009). Also emitted from the sun are the streams of electrons and protons which comprise the background solar wind and impulsive fluxes of energetic charged particles contained in solar energetic particle (SEP) events (Parker, 1958; Ryan et al., 2000). Charged particle gases from the sun are classified as high-beta plasmas within which the remnants of solar magnetic fields are transported towards Earth and can interact with the geomagnetic field. The transported field is referred to as the Interplanetary Magnetic Field (IMF) and the orientation of this field relative to Earth has a controlling effect on the degree of coupling of the solar wind and/or the transient streams with the magnetosphere; that is, the earth's outer magnetic shielding layer that acts to protect the terrestrial biosphere from interplanetary energetic charged particles (Schatten, 1971; Yermolaev et al., 2018). However as the solar streams, with their embedded magnetic fields, impact the magnetosphere they can enhance geomagnetic activity which, in turn, can have a significant effect on the coupled Magnetosphere-Ionosphere-Thermosphere (MIT) system. For example, within interplanetary space, a solar High-Speed Stream (HSS) can overtake a preceding Low-Speed Stream (LSS) thus forming a dense Corotating Interaction Region (CIR) (Gosling and Pizzo, 1999) that can profoundly increase the level of geomagnetic activity. The sun also periodically releases large-scale "clouds" of plasma in





the form of a Coronal Mass Ejection (CMEs), which, when propagated into interplanetary space, is termed an Interplanetary CME (ICME) (Gosling et al., 1990). When directed towards earth, CIRs and ICMEs can initiate geomagnetic storms resulting

in large-scale perturbations of the MIT system lasting up to several days (Borovsky and Denton, 2006). Flares represent another class of transient solar phenomena which can affect the MIT system. A solar flare is a large scale (on solar dimensions) reconfiguration of the photospheric magnetic field resulting in the impulsive release of vast amounts of energy and a redistribution of solar mass (Philips, 1991). Electromagnetic radiation and extremely energetic (relativistic) particles released during a solar flare event can result in an abrupt increase in the ionospheric density near the subsolar point and within the high-latitude polar

caps (Sauer and Wilkinson, 2008). A Sudden Ionospheric Disturbance (SID) is the result of the increased solar UltraViolet (UV) and X-ray radiative flux released in solar flares (Mitra, 1974) whereas a Polar Cap Absorption (PCA) event is the result of energetic particles entering the atmosphere along "open" magnetic field lines connect to interplanetary space (Rose and Ziauddin, 1962).

The systematic monitoring of sunspots over the last two centuries has shown that solar activity exhibits an approximate 11-year temporal cycle during which the observed SunSpot Number (SSN) (Clette et al., 2014) varies from a local solar minimum near 0; that is, no spots observed, to a solar maximum of up to several hundred spots visible on the solar disk. Near solar maximum the total radiant energy from the sun reaches a corresponding peak along with a propensity for short-lived solar transients of increased radiation and particle emissions. These solar transients are the main drivers of space weather (Song et

al., 2001; Knipp, 2011). Figure 2 (after illustrates this cyclic variation in the monthly-averaged SSN along with the related solar-geophysical indices for the solar radio flux (F10.7) and the geomagnetic Ap. The F10.7 index (Tapping, 2013) is a local noontime measurement of the solar radio flux at a wavelength of 10.7 cm, corresponding to a radio-wave frequency of 1400 MHz. The F10.7 index is often used as a proxy for upper atmospheric heating from solar Extreme UltraViolet (EUV) radiation. The F10.7 index is given in solar flux units (sfu; 1 sfu = $10^{-22}$ Wm$^{-2}$ Hz$^{-1}$) and typically ranges from <50 sfu at solar

minimum to >300 sfu at solar maximum. The daily Ap index (Rostoker, 1972) is derived from the 3-hour Kp index from which an additional Joule heating effect associated with geomagnetic activity can be estimated. The Ap index is provided in units of nanoTeslas (nT) and typically ranges from ∼5 nT (solar minimum) to ∼40 nT (solar maximum). Near the time of solar maximum the increased frequency of solar transients; i.e. flares and CMEs, can lead to increased geomagnetic activity. As energy inputs to the MIT system, the increased solar radiation and enhanced geomagnetic activity heat the thermosphere

and cause the atmosphere to diffuse outward from lower altitude regions of higher neutral gas pressures (densities) to the more tenuous upper atmosphere. The consequential increase in atmospheric drag associated with a more dense atmosphere affects the motion of a LEO satellite and expends energy at the expense of the orbit. The monthly smoothed values for F10.7 and Ap, as plotted in Figure 2, tend to obscure the effects of solar events and geomagnetic transients. The impacts of increased solar-geomagnetic activity on the atmosphere and, in turn, the atmospheric drag are discussed in Nwankwo et al. (2015); and

references therein.

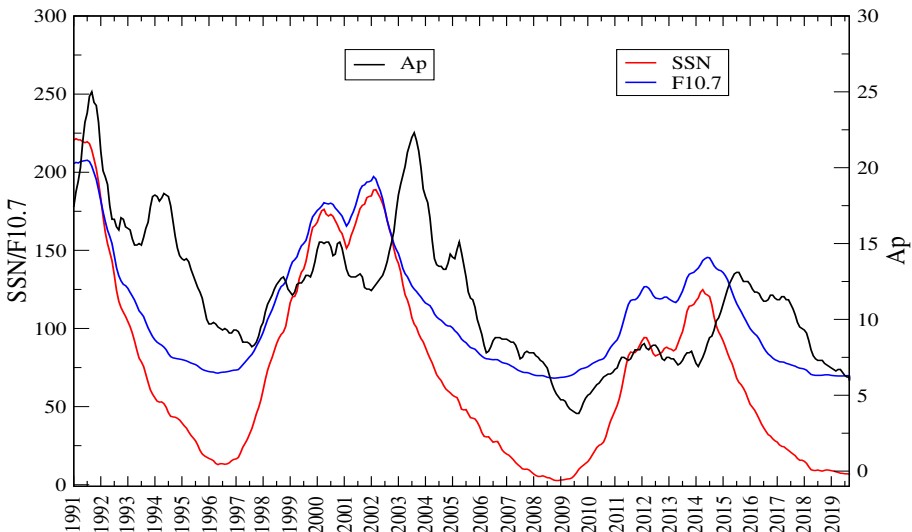

**Figure 2.** Monthly smoothed values of archived Sun Spot Number (SSN), Radio solar flux (F10.7) and geomagnetic Ap from January 1991 - March 2020 (updated predicted data in Nwankwo and Chakrabarti (2013) with actual data).

## 2.1 Relevance of the study and its application

Rapid variations in the local thermospheric density increase the risks of satellite collisions due to larger error margins in space-craft positioning and motion. In 2009 a Russian satellite in orbit (Cosmos 2251) collided with a United States communications satellite (Iridium 33) at an altitude of about 800 km (Jakhu, 2009; Kelso, 2009). Cosmos 2251 was a defunct satellite in orbit
whereas Iridium 33 was an operational satellite providing telecommunication services when the accident occurred. In addition to the total destruction of the satellites, this hyper-velocity collision resulted in a large increase in the amount of small, but still potentially lethal, space debris. Assessing atmospheric drag-associated risk is imperative due to the increasing number of both active and expired space missions combined with a less than fully specified debris field (McCrea, private com., 2018). For example, the planned launch of new capabilities, such as SpaceX's Starlink Mega Constellation makes this subject increasingly
germane to satellite operators and stakeholders. Space agencies acknowledge the potential threat posed by solar-geomagnetic activity in modulating satellite trajectories and are making strides towards addressing the issue. For example, the European Space Agency (ESA) is currently assessing space-weather related risks within the framework of its Space Situational Awareness (SSA) program (Bobrinsky and Del Monte, 2010). An important mitigation approach (among others) to safeguard satellite operations is the development and implementation of models that can assess the impact of space weather on LEO satellite track-
ing (Nwankwo et al., 2019). Accordingly, this work is of practical importance as the resulting model and simulation support efforts to increase SSA and improve collision risk mitigation.





## 2.2 Data, method and scope of the study

In this work we have modeled LEO satellite trajectories during intervals of disturbed and quiet solar-geomagnetic conditions to better understand how space weather affects atmospheric drag and, in turn, satellite orbits. The model is applied to two
satellites with different ballistic coefficients as detailed in Table 1. Sat-A and Sat-B represent typical SmallSats of mass, $m_s$, and projected area, $A_s$. The selected "real-world" intervals were chosen based of a review of the environmental parameters that describe solar-geomagnetic activity including the solar wind speed ($V_{sw}$) and proton density (PD), the disturbance storm time (Dst) index (Mayaud, 1980), the IMF $B_y$ and $B_z$ components, and the auroral electrojet (AE) index (Davis and Sugiura, 1966). The interplanetary parameters ($V_{sw}$, PD, IMF $B_y$ and $B_z$) and the geomagnetic responses in Dst and AE are reflective of the
processes by which energy is transferred from the solar wind to the MIT system (Nwankwo (2016); and references therein). Model runs of the atmospheric density profile were made for the quiet environmental interval of July 2006 and for the disturbed conditions of July 2000.

## 2.3 Quiet environmental conditions near solar minimum (1-31 July 2006)

Solar cycle 23 was on its descending phase in 2006 heading towards a solar minimum in December 2008. Solar minimum is
usually accompanied by a reduction in both solar radiant emissions and the frequency of solar transient events. The monthly averages of F10.7 and Ap for July 2006 were 78.4 sfu and 6.5 nT, respectively. Figure 3 is a plot of the 1-hour averaged variations in $V_sw$, PD, Dst, IMF $B_y$ and $B_z$ and AE for July 2006. The most notable feature, or lack thereof, was the essentially flat Dst index throughout the month that is indicative of no significant geomagnetic storms. However there were a number of interesting features related to the state of the solar-terrestrial environment. In particular, we note the character of the background
solar wind speed, $V_sw$, and density, PD, on 04 July which is indicative of a pressure buildup on the nose of the magnetopause. These data are suggestive of a CIR that was not particularly well coupled to the magnetosphere due to a non-favourable IMF $B_z$ (Pokhotelov et al., 2009). Supporting evidence of a CIR was the simultaneous detection of an increased flux of energetic protons (data not included here) observed just inside the magnetopause by the Geostationary Operational Environmental Satellite (GOES)(Posner et al., 1999). Similar "CIR-like" features in Figure 3 were the interplanetary parameters for 28 July and 31 July
although their strengths were apparently less intense than the feature observed earlier in the month. We suspect that the sources for these CIRs were high-stream flows originating from within solar Coronal Holes (CHs). A review of solar imagery (data not included here) available from the Extreme ultraviolet Imaging Telescope (EIT) on the Solar and Heliospheric Observatory (SOHO) satellite revealed that there was a distinctly visible CH that crossed the solar disk in early July. This CH may well have been the source of a high-stream flow that resulted in the detected CIR on 04 July. During the middle of the month no CHs
were apparent excepting a stationary, non-geoeffective, polar CH in the northern solar hemisphere. However, later in the month several CHs could be seen on the solar disk which might have been the sources for the solar wind features detected on 28 July and 31 July. The final characteristic of interest in Figure 3 was the periodic geomagnetic activity observed in the AE substorm index which was suggestive of High-Intensity, Long-Duration, Continuous AE Activity (HILDCAA). To this end, Guarnieri et al. (2006) noted that HILDCAA events can often be associated with CIRs, particularly on the downside of the solar cycle,





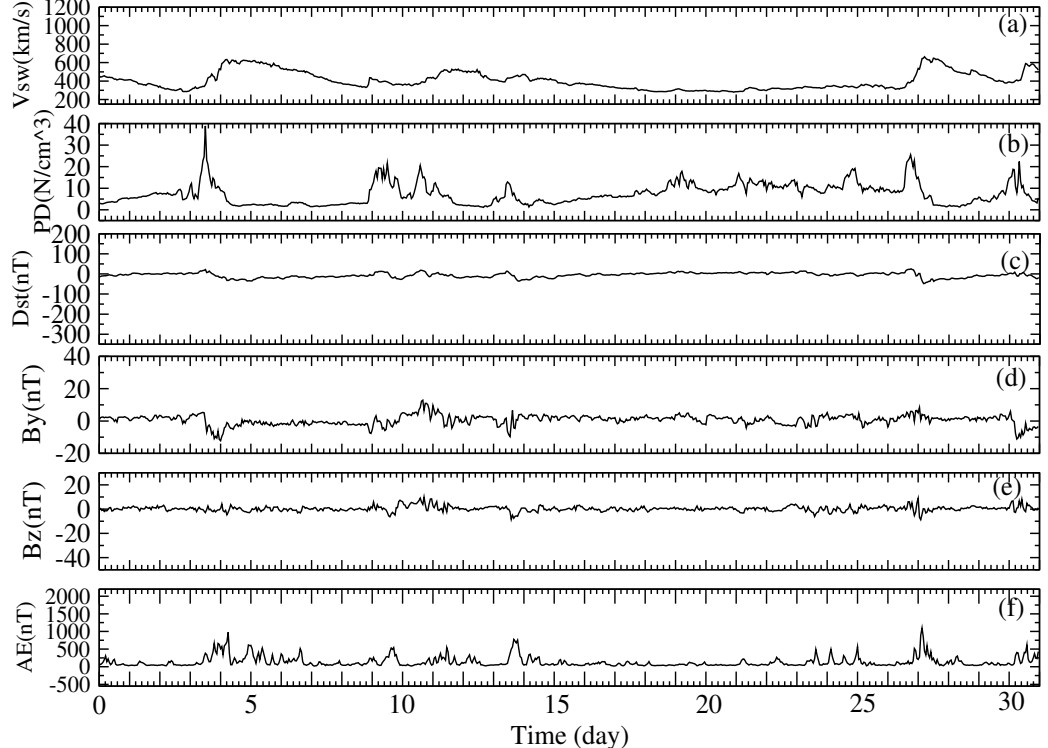

**Figure 3.** 1-hour averaged variations in Vsw, PD, Dst, IMF By and Bz, and AE for the geomagnetically quiet, solar minimum interval 1-31 July 2006.

as was the case here. The atmospheric drag effects modeled for this solar minimum interval of relative quiet will be compared and contrasted to the disturbed period of July 2000.

### 2.3.1 Disturbed environmental conditions near solar maximum (1-31 July 2000)

Year 2000 witnessed the expected rise in overall solar activity as the sun was progressing towards the maximum of cycle 23 which peaked in November 2001. Figure 4 is a plot of the hourly-averaged interplanetary and geomagnetic parameters for July 2000. The related monthly averages of the F10.7 and Ap indices were 212.2 sfu and 21.4 nT, respectively. Germane to this interval were the solar wind drivers and, more importantly, their significant fluctuations and increases, in the PD on days 01-04 July, 09-15 July and 25-29 July. These fluctuations had significant consequences on the MIT system. Of note was the occurrence on 15 July of an intense geomagnetic storm (Gonzalez et al., 1994) having a Dst of -301 nT. The apparent source of this, so-called Bastille Day event which nominally spanned the 3-day interval 14-to-16 July, was a geoeffective CME that was first observed erupting from the sun at 10:54 UT on 14 July in association with an X5.7 flare within active region #19077 at solar location N22W07 (Denig et al., 2018). On 15 July a large Sudden Storm Commencement (SSC) of 112 nT



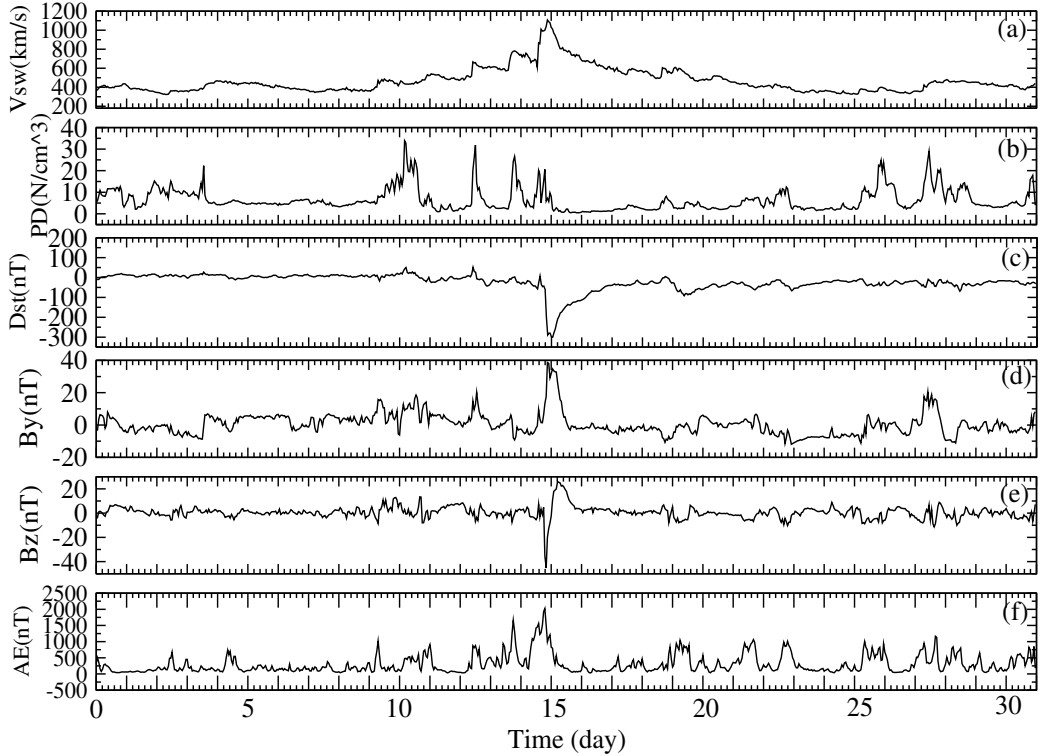

**Figure 4.** 1-hour averaged variations in Vsw, PD, Dst, IMF By and Bz, and AE for the geomagnetically quiet, solar minimum interval 1-31 July 2000.

at 14:37 UT marked the arrival of the CME at the magnetopause and the start of the geomagnetic storm main phase (Closs, 1967). Given a total transit time of just under 28 hours the estimated CME speed from the sun to earth was a fast ∼1500 km/s which is consistent with an initially observed speed of 1673 km/s (Denig et al., 2018) and assessments of the probability of a significant geomagnetic response (Srivastava and Venkatakrishnan, 2002). The 40-nT magnitude and initial negative polarity of the IMF $B_z$ suggests that significant energy was transferred to the MIT system from the solar wind. The related increase in AE corresponding to a substorm occurring within the geomagnetic storm (McPherron et al., 1973; Kepko et al., 2015) indicates enhanced ionospheric currents within the auroral zone due to the strong coupling between the IMF and the MIT (Pudovkin et al., 1995). Clearly, the upper atmosphere was significantly disturbed throughout July 2000 and, in particular, during to the Bastille Day event. The expected consequences of the enhanced solar-geomagnetic activity for July 2000 was increased atmospheric drag and a consequential decrease in the satellite orbital altitude.





## 3 Modeling atmospheric drag effect on LEO satellites' trajectory

We previously formulated and solved a set of coupled differential equations to obtain the instantaneous position, velocity and acceleration of a typical LEO satellite under the influence of atmospheric drag (Nwankwo and Chakrabarti, 2014, 2015; Nwankwo et al., 2015; Nwankwo, 2018). A spherical coordinate system $(r, \theta, \phi)$ was used with an origin at the center of the earth and an assumed constant polar angle; that is, $\theta = constant$. In satellite parlance a constant polar angle is equivalent, in principle, to a constant satellite inclination angle. Orbital decay was determined as a consequence of changes in the radial distance, $r$, and the azimuthal angle, $\phi$, through the following set of coupled equations:

$$\dot{v}_r = -\dot{\phi} r^2 \frac{A_s C_d}{m_s}, \tag{1}$$

$$\dot{r} = v_r, \tag{2}$$

$$\ddot{\phi} = -\frac{1}{2} r \rho \dot{\phi}^2 \frac{A_s C_d}{m_s}, \tag{3}$$

$$\dot{\phi} = \frac{v_\phi}{r}, \tag{4}$$

where $v_r$ and $v_\phi$ are, respectively, the radial and tangential velocity components and are, respectively, the azimuthal angular velocity and azimuthal angular acceleration. The parameters $C_d$, $A_s$, and $m_s$ were defined in Section 1.0 - recall that the expression $C_d A_s / m_s = B$ is the ballistic coefficient. In the current analysis, the radial velocity, $v_r$, is used to calculate the daily ODR whereas the radial distance, $r$, is used to model changes in satellite altitude.

**Table 1.** Orbital and ballistic parameters used in this study

| Satellite | Altitude (km) | $m_s$ (kg) | $A_s$ (m$^2$) | $C_d$ | B (m$^2$/kg) |
|---|---|---|---|---|---|
| Sat-A | 450 | 250 | 0.25 | 2.2 | $2.200 \times 10^{-3}$ |
| Sat-B | 450 | 522 | 0.72 | 2.2 | $3.034 \times 10^{-3}$ |

### 3.1 Atmospheric density model

The effects of atmospheric drag on LEO satellites and hence the rates at which satellite orbits decay largely depend on the atmospheric density which, in turn, is heavily influenced by solar and geomagnetic activity (Fujiwara et al., 2009). Accurate



knowledge of atmospheric drag requires a high-fidelity model of the in-situ neutral-gas density or, more generally, the atmospheric density profile. Supporting information regarding the level of atmospheric heating and, in turn, atmospheric expansion can be gleaned from knowledge of the atmospheric temperature profile. The upper atmosphere, or thermosphere, exhibits large solar-cycle variations in temperature, density, composition and winds (Walterscheid, 1989). A number of high-quality models are available that provide suitable approximations of atmospheric profiles of density, $\rho$, and temperature, T (Picone et

al., 2002; Bruinsma et al., 2003; Bowman et al., 2008). For this work we have selected the Naval Research Laboratory Mass Spectrometry and Incoherent Scatter Extended 2000 (NRLMSISE-00) empirical atmospheric model. NRLMSISE-00 consists of parametric and analytic approximations to physical theory for the vertical structure of the atmosphere as a function of time, location, solar and geomagnetic activity. While a global specification was used to extract the density along the satellite flight path, the atmospheric curves used in Figures 5, 7 and 11 (to be discussed) to represent a general atmospheric response used a

reference altitude of 450 km. The solar-geomagnetic parameters used for NRLMSISE-00 model are the daily values of F10.7 and Ap (Nwankwo and Chakrabarti (2018); and references therein).

## 4    Results and discussion

The results of our simulation arising as solutions to the above set of coupled differential equations are presented in this section. The environmentally quiet interval of July 2006 is presented as the baseline for atmospheric drag whereas the disturbed interval

of July 2000 illustrates the deleterious impact that solar-geomagnetic activity can have on satellite orbits. Within each of these intervals a 12-day period of environmentally quiet and exceptionally disturbed activity, respectively, is used to highlight the impact of extreme conditions.

### 4.1    Atmospheric drag effects for quiet solar-geomagnetic activity (July 2006)

Figure 5 depicts the mean daily variations in Dst, Bz, atmospheric density ($\rho$) and temperature (T), and the altitude (h) and

orbit decay rates (ODRs) of Sat-A and Sat-B under the quiet solar-geomagnetic conditions of July 2006. During this 1-month interval of relatively low environmental stress, the ranges of Dst and Bz (daily mean) are -28.71−7.75 (nT) and -1.18−1.84 (nT), respectively. T varied between 770 °K and 880 °K, whereas $\rho$ varied between $0.33 \times 10^{-12}$ kg/m$^3$ and $0.55 \times 10^{-12}$ kg/m$^3$. These atmospheric parameters are consistent with low geomagnetic activity, solar minimum conditions (Fujiwara et al., 2009). The orbital drag calculations indicate that Sat-A decayed by 0.52 km during the month with an ODR ranging from 13-to-23

m/day whereas Sat-B decayed by 0.65 km with an ODR range of 16-to-29 m/day. These modest yet consistent differences in ODR and decay for Sat-A and Sat-B reflect the differences in their respective ballistic coefficients (see Table 1). Figure 6 is a plot of the daily F10.7, SSN and Ap indices. While no significant geomagnetic storms occurred during the entire month we note that the minor increases in the daily Ap for 05 July, 28 July and possibly 31 July corresponded to slight increases in the atmospheric parameters and the peak ODRs of 23 m/day (29 m/day) for Sat-A (Sat-B). The baseline ODRs for July 2006 will

be contrasted with the model decay rates for the solar maximum, geomagnetically disturbed interval of July 2000.





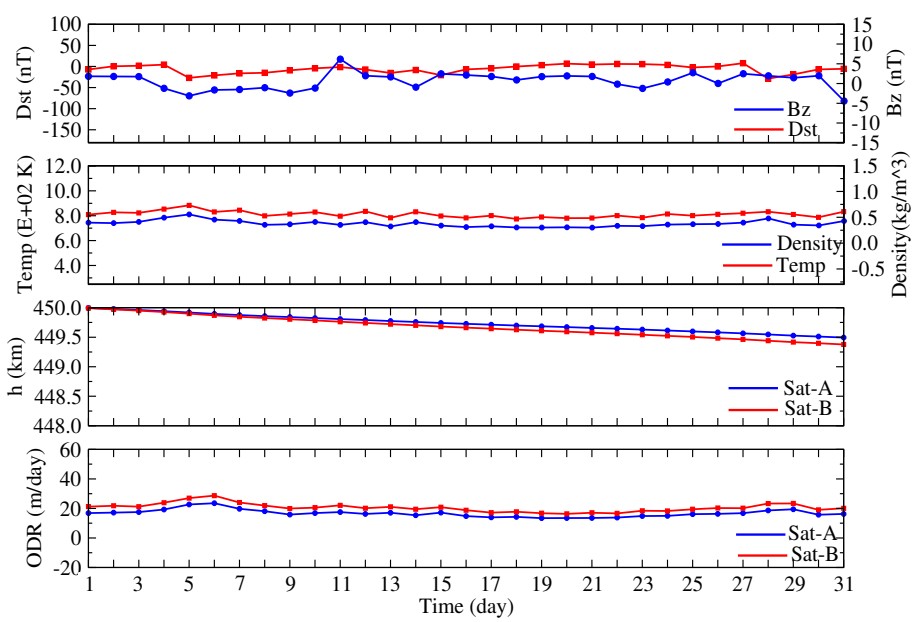

**Figure 5.** Mean daily variations in Dst, Bz, ρ, T, h and ODR for Sat-A (blue) and Sat-B (red) for quiet conditions (July 2006).

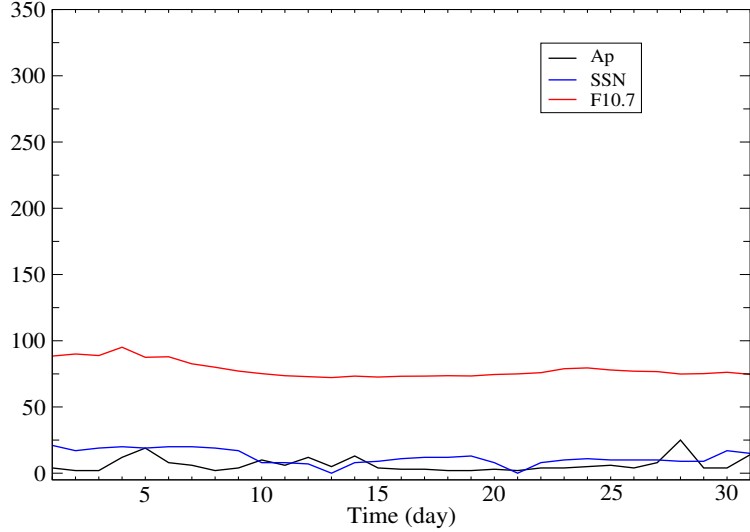

**Figure 6.** Daily values of Ap, International SSN and F10.7 for July 2006.

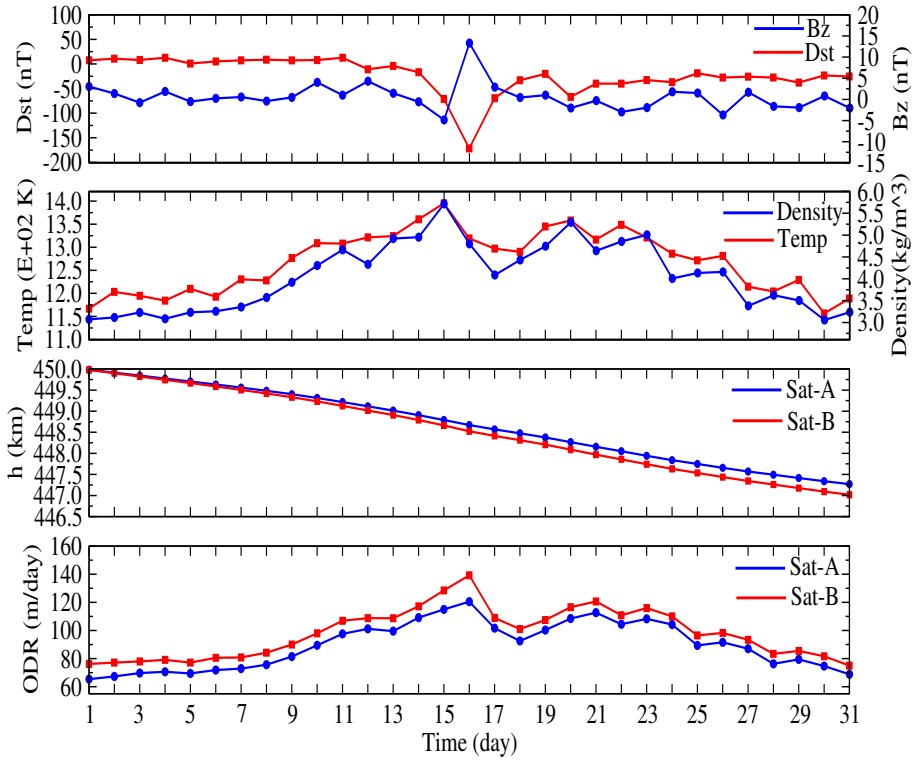

**Figure 7.** Mean daily variations in Dst, Bz, ρ, T, h and ODR for Sat-A (blue) and Sat-B (red) for quiet conditions (July 2000).

## 4.2 Atmospheric drag effects for enhanced solar-geomagnetic activity (July 2000)

Figure 7 depicts the mean daily variations in Dst, Bz, ρ, T, h, ODR for the environmentally enhanced interval of July 2000. During this month the range of daily values of Dst and Bz are -171.63−12.75 (nT) and -4.84−13.30 (nT), respectively. The modeled temperatures for the month varied from 1156 °K to 1580 °K which were indicative of a generally warm atmosphere
near solar maximum (Fujiwara et al., 2009). In response, the thermospheric densities for an expanded atmosphere ranged from $2.4 \times 10^{-12}$ kg/m$^3$ to $5.7 \times 10^{-12}$ kg/m$^3$ (Fujiwara et al., 2009; Emmert, 2015). Accordingly, Sat-A decayed by about 2.77 km whereas Sat-B decayed by about 3.09 km. Again, as expected, the integrated effect of atmospheric drag on Sat-B was greater than that experienced by Sat-A due to differences in the ballistic coefficients. The range of daily values for Sat-A's ODR was 65 m/day to 120 m/day whereas the range of ODR for Sat-B was 78 m/day to 142 m/day. An interesting trend is the
general increase then decrease in the ODR throughout the month which is consistent with the variations in the modeled ρ and T.

In order to further investigate this trend we plotted in figure 8 the daily indices for F10.7, SSN and Ap and found that the density and temperature trends were consistent with the solar F10.7 and SSN indices. Again, this is not surprising in that F10.7 is a key input to the NRLMSISE-00 model. The apparent similarity in the behaviors of the daily SSN and daily F10.7 index was

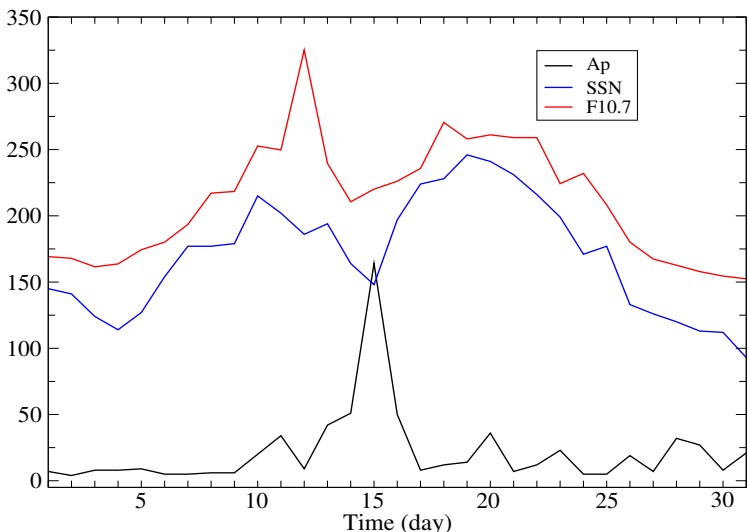

**Figure 8.** Daily values of Ap, International SSN and F10.7 for July 2000.

also reasonable given that intense radiant emissions from solar faculae are proportional to the number of sunspots (Ambelu et al., 2011). However, we note that the significant spike in F10.7 (due to the intense flare) did not reflect in the simulated ODR. This outcome is consistent with the findings of Knowles et al. (2001) who stated that "the effect of geomagnetic activity appears to be more important than that of prompt radiation. The model values for ODC, as well as the atmospheric density and temperature, spiked predictably mid-month in response to the additional energy input from the Bastille Day geomagnetic

storm. The thermospheric T and $\rho$ on the peak storm day (16 July) were 1580 $^\circ$K and $5.7 \times 10^{-12}$ kg/m$^3$, and the corresponding values of ODRs for Sat-A and Sat-B were 120 m/day and 142 m/day, respectively. A more detailed plot of the 3-hour magnetic a$_p$ index included in figure 9 for 13-17 July indicates that the geomagnetic storm lasted about one day, starting near noon on 15 July and ending some 24 hours later. We note that the start of the storm was consistent with the previously mentioned SSC that occurred at 14:37 UT on 15 July. Also occurring during the month of July 2000 were a series of minor disturbances (days

11, 20, 23, 26 and 28-29 (Fig. 8)) which contributed to the enhanced temperatures and densities (beyond solar inputs alone) observed throughout the month.

The results of our simulation showed that the maximum ODRs for Sat-A and Sat-B were in response to the Bastille Day event. We glean from these results that the background atmospheric parameters were responsive to both the slowly-varying solar irradiance inputs during the month and to the impulsive geomagnetic storm inputs. These results confirm that the transient

response of satellite drag to impulsive geomagnetic storms lead to the largest uncertainties in orbit dynamics and pose a great risk to efficient satellite operations.





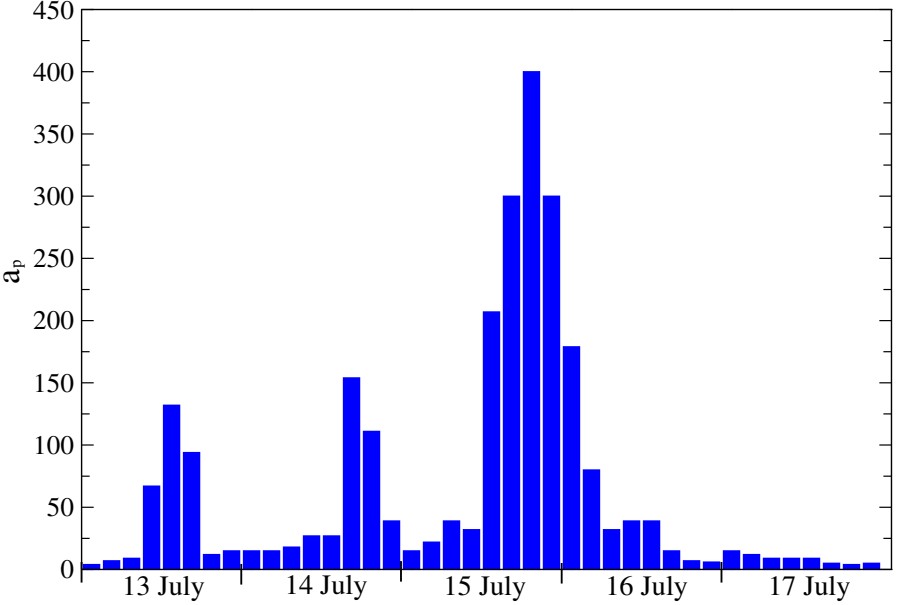

**Figure 9.** Three-hour $A_p$ values for 13-17 July 2000.

### 4.3 Intervals of exceptionally quiet and disturbed environmental stress

In this section we focus on the 12-day sub-intervals of elevated solar and geomagnetic activity for 09-20 July 2000 and of exceptionally quiet activity for 15-26 July 2006. For these intervals we compare and contrast the activity levels on the satellite

trajectory parameters (h, ODR) with the corresponding perturbations in T and $\rho$. Figure 10 depicts 1-hour averaged variations in $V_{sw}$, PD, Dst, IMF $B_y$ and $B_z$, and AE for the intervals of high activity (left) and low activity (right). In Fig 11 we show the corresponding daily variations in Dst, Bz, $\rho$, T, h and ODR for Sat-A with the lower ballistic coefficient (blue trace) and Sat-B with the higher ballistic coefficient (red trace) within the intervals of elevated activity (left) and quiescent activity (right). During the sub-interval of elevated activity (left) the Sat-A orbit decayed by 1.14 km and the ODR increased from 81.46 m/day

on 09 July to 120 m/day on 16 July which is just after the peak of the Bastille Day storm. Similarly, Sat-B decayed by about 1.27 km, and the ODR increased from 91.85 m/day on 09 July to 142 m/day on 16 July. Considering the sub-interval of low activity (right) Sat-A (blue trace) decayed by a total of 0.16 km with an ODR ranging from 13.41 to 17.17 m/day whereas Sat-B (red trace) decayed by a total of 0.20 km with an ODR ranging from 16.67 to 21.25 m/day. The salient features for the orbital decay and maximum ODR for both satellites for the active versus quiet conditions are summarized in Table 2. The stark

contrast between the two regimes indicates that solar-geomagnetic activity had a more than 6-fold (7-fold in this case) impact on the orbital parameters for the modeled conditions. This dramatic effect makes it imperative that the orbital parameters for relevant space objects in LEO be frequently updated (Knowles et al., 2001).





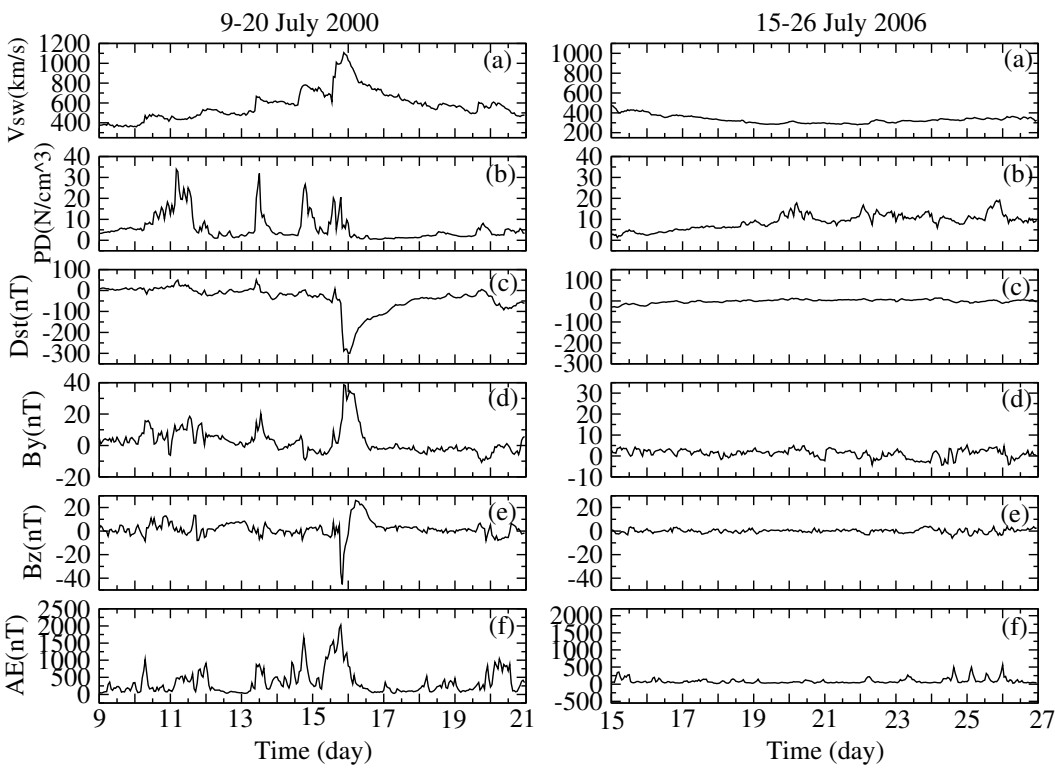

**Figure 10.** 1-hour averaged values of Vsw, PD, Dst, IMF By and Bz, and AE during 9-20 July 2000 (left) and 15-26 July 2006 (right) **(Nwankwo et al., 2020)**

**Table 2.** Summary of altitude decay and ODR of Sat-A and Sat-B for the extreme (12-day) activity levels

| Date Range | Activity Level | Satellite | B (m$^2$/kg) | Decay (kg) | Max ODR (m/day) |
|---|---|---|---|---|---|
| 09-20 July 2000 | High | Sat-A | $2.200\times10^{-3}$ | 1.14 | 120.0 |
| 09-20 July 2000 | High | Sat-B | $3.034\times10^{-3}$ | 1.27 | 120.0 |
| 15-26 July 2000 | Low | Sat-A | $2.200\times10^{-3}$ | 0.16 | 17.2 |
| 15-26 July 2000 | Low | Sat-B | $3.034\times10^{-3}$ | 0.20 | 21.2 |

In Figure 12, we show detailed analysis (and comparison) of how h and ODR of the two satellites varied during the regimes of elevated and exceptionally quiet solar and geomagnetic activity. The goal of this analysis is to demonstrate how enhanced
atmospheric drag caused by the July 2000 Bastille Day event affected LEO satellites in contrast to the interval of exceptionally quiet geomagnetically activity conditions during 15-26 July 2006. We describe new indices in Tables 3 and 4 for the analysis. Associating the Tables with the geometry of curves in figure 12 gives a better understanding of the analysis to follow. When compared with the 12-day interval of exceptionally quiet geomagnetic activity, the total decay (h) and ODR increase

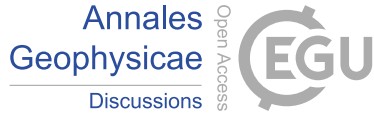



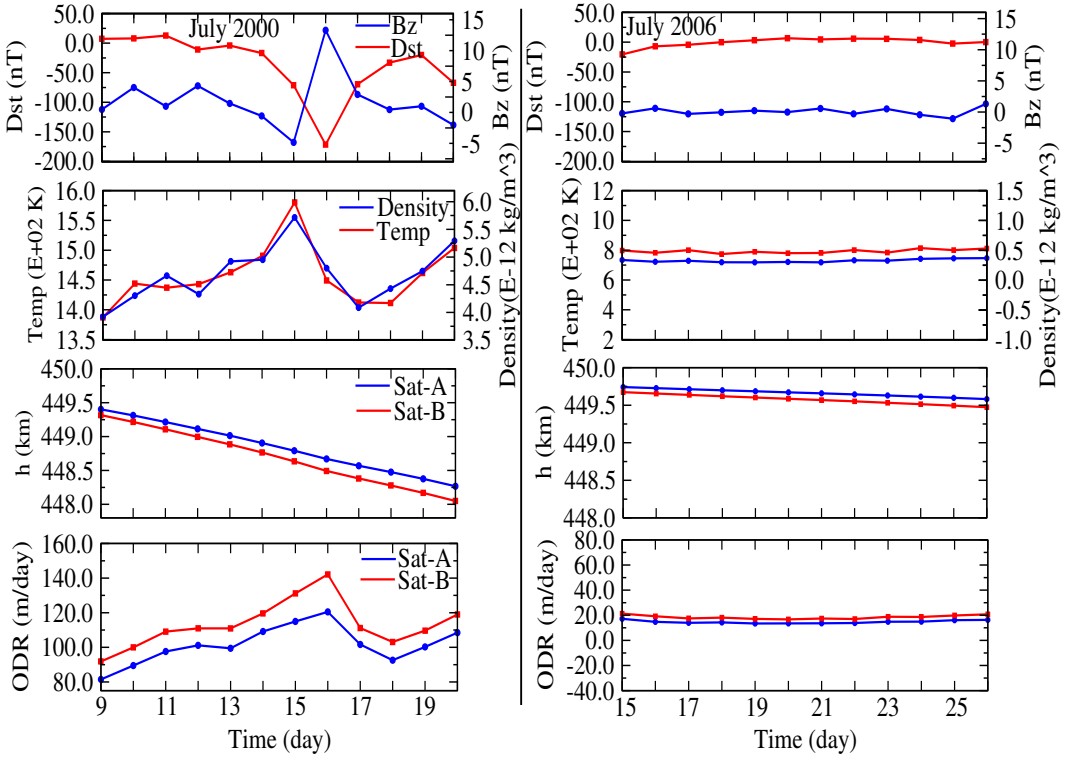

**Figure 11.** Corresponding mean daily variations in Dst, Bz, $\rho$, T, h and ODR for Sat-A and Sat-B during 09-20 July 2000 (left) and 15-26 July 2006 (right)

(from the background or initial values) during the elevated geomagnetic activity are 0.98 km and 80.24 m/day for Sat-A (de-

scribed by $dh_{Sat-A}$ and $\triangle ODR_{Sat-A}$, respectively), and 1.07 km and 84.12 m/day for Sat-B (described by $dh_{Sat-B}$ and $\triangle ODR_{Sat-B}$). The decay caused by the Bastille Day event are 0.34 km and 0.39 km for Sat-A and Sat-B, respectively (described by $h_{BD-A}$ and $h_{BD-B}$). The respective ODR increments (due to Bastille Day event) are 40.23 m/day and 58.00 m/day (described by $\triangle ODR_{BD-A}$ and $\triangle ODR_{BD-B}$). The corresponding percentage increase for the two parameters (h and ODR) when contrasted with the interval of exceptionally quiet geomagnetic activity are 74.24% and 66.74% for Sat-A, and 75.35%

and 59.18% for Sat-B. However, the additional percentage increase of Bastille Day event to the monthly mean values of July 2000 are 34.69% (h) and 50.13% (ODR) for Sat-A, and 36.45% and 68.95% for Sat-B. This result implies that storms of this magnitude can add more than 30% and 50-70% increase to background h and ODR during the interval.



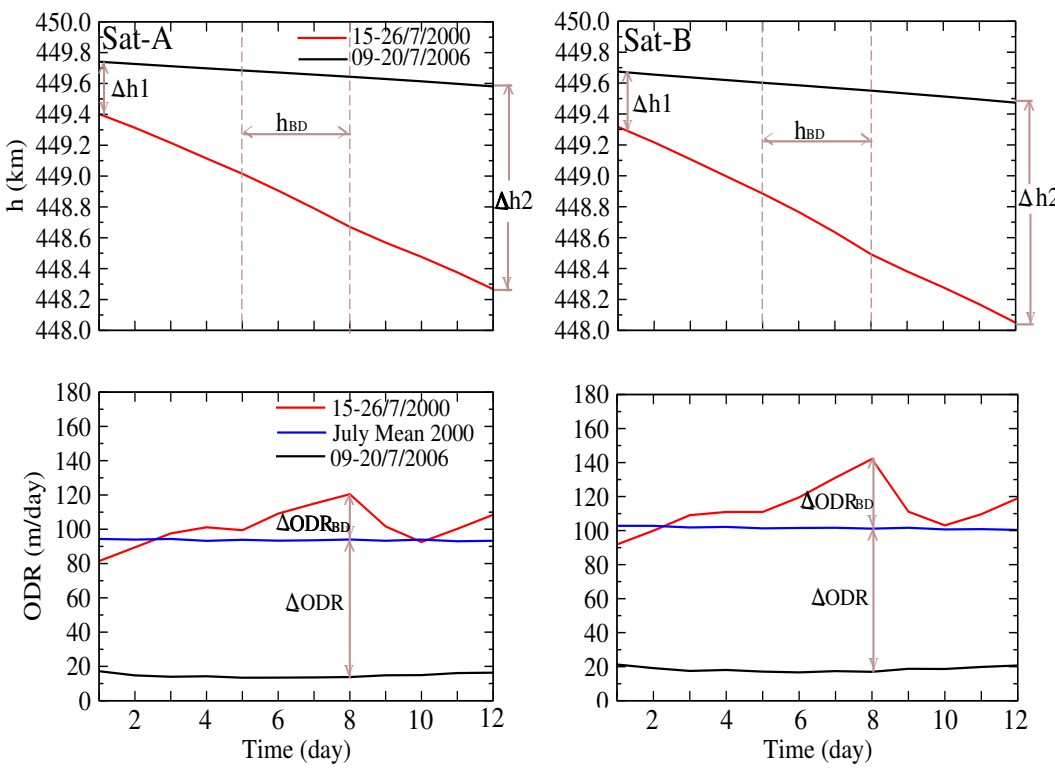

**Figure 12.** Contrast between the daily variations in h and ODR of Sat-A (left) and Sat-B (right) during 09-20 July 2000 and 15-26 July 2006

## 5 Conclusions

Solar activity in the form of increased solar irradiance and flux of energetic particles form important channels through which the earth's atmosphere is impacted. Atmospheric heating and expansion can significantly increase orbital drag which, in turn, perturbs satellite trajectories and results in accelerated orbital decay. In this work, we simulated the effect of atmospheric drag on two hypothetical SmallSats in LEO with different ballistic coefficients during 1-month long intervals of disturbed and quiet solar-geomagnetic activity. During a 1-month period of enhanced activity (01-31 July 2000) the increased density of the upper

atmosphere caused a modeled mean decay of 2.77 km (3.09 km) for the satellite with the smaller (larger) ballistic coefficient. Conversely, for the more quiescent period (01-31 July 2006) the mean decay was only 0.52 km (0.65 km) for the respective satellites. Further analysis and simulation of atmospheric drag for periods of elevated (or extreme) solar-geomagnetic activity during 09-15 July 2000 and exceptionally quiet geomagnetic activity (15-26 July 2006) resulted in Sat-A (Sat-B) modeled orbital decays of 1.14 km (1.27 km) and 0.16 km (0.20 km), respectively. We also estimated the enhanced atmospheric drag

effect on the satellites's parameters caused by the July 2000 Bastille Day event in contrast to the interval of geomagnetically quiet conditions. While the percentage increase of $h$ and ODR due to elevated geomagnetic activity (of 9-20 July 2000) are 74.24% and 66.74% (75.35% and 59.18%) for Sat-A (Sat-B), the additional (daily) percentage increase due to the Bastille Day





**Table 3.** Indices used for description of the intervals of elevated and exceptionally quiet geomagnetic activity. EQGA connote exceptionally quiet geomagnetic activity

| s/n | Abbrev/Symbol | Definition | Value |
|---|---|---|---|
| 1 | $h1_{A06}$ | Sat-A's height on 15 July 2006 | 449.74 km |
| 2 | $h2_{A06}$ | Sat-A's height on 26 July 2006 | 449.58 km |
| 3 | $h1_{A00}$ | Sat-A's height on 9 July 2000 | 449.40 km |
| 4 | $h2_{A00}$ | Sat-A's height on 20 July 2000 | 448.28 km |
| 5 | $h1_{B06}$ | Sat-B's height on 15 July 2006 | 449.67 km |
| 6 | $h2_{B06}$ | Sat-B's height on 26 July 2006 | 449.47 km |
| 7 | $h1_{A00}$ | Sat-A's height on 9 July 2000 | 449.32 km |
| 8 | $h1_{B00}$ | Sat-B's height on 20 July 2000 | 448.05 km |
| 9 | ODR$_{A06}$ | orbit decay rate of Sat-A during EQGA corresponding to 15 July 2000 | 13.79 m/day |
| 10 | ODR$_{B06}$ | orbit decay rate of Sat-B during EQGA corresponding to 15 July 2000 | 17.00 m/day |
| 11 | ODR$_{A\bar{0}\bar{0}}$ | mean orbit decay rate of Sat-A for July 2000 | 94.03 m/day |
| 12 | ODR$_{B\bar{0}\bar{0}}$ | mean orbit decay rate of Sat-B for July 2000 | 101.14 m/day |
| 13 | ODR$_{BD-A}$ | orbit decay rate value of Sat-A on Bastille Day | 120.47 m/day |
| 14 | ODR$_{BD-B}$ | orbit decay rate value of Sat-B on Bastille Day | 142.12 m/day |

**Table 4.** Indices used for analysis of the intervals of elevated and exceptionally quiet geomagnetic activity (EEQGA) and the Bastille Day event (BDE).

| s/n | Abbrev/Symbol | Definition | Value |
|---|---|---|---|
| 1 | $\triangle h1_{Sat-A}$ | $h1_{A06} - h1_{A00}$ | 0.34 km |
| 2 | $\triangle h2_{Sat-A}$ | $h2_{A06} - h2_{A00}$ | 1.32 km |
| 3 | $dh_{Sat-A}$ | $\triangle h2_{Sat-A} - \triangle h1_{Sat-A}$ | 0.98 km |
| 4 | $\triangle h1_{Sat-B}$ | $h1_{B06} - h1_{B00}$ | 0.36 km |
| 5 | $\triangle h2_{Sat-B}$ | $h2_{B06} - h2_{B00}$ | 1.42 km |
| 6 | $dh_{Sat-B}$ | $\triangle h2_{Sat-B} - \triangle h1_{Sat-B}$ | 1.07 km |
| 7 | $h_{BD-A}$ | $h_{Sat-A}$ (13 July 2000) $- h_{Sat-A}$ (16 July 2000) | 0.34 km |
| 8 | $h_{BD-B}$ | $h_{Sat-B}$ (13 July 2000) $- h_{Sat-B}$ (16 July 2000) | 0.39 km |
| 9 | $\triangle$ODR$_{Sat-A}$ | ODR$_{A\bar{0}\bar{0}} -$ ODR$_{A06}$ | 80.24 m/day |
| 10 | $\triangle$ODR$_{BD-A}$ | ODR$_{BD-A} - \triangle$ODR$_{Sat-A}$ | 40.23 m/day |
| 11 | $\triangle$ODR$_{Sat-B}$ | ODR$_{B\bar{0}\bar{0}} -$ ODR$_{B06}$ | 84.12 m/day |
| 12 | $\triangle$ODR$_{BD-B}$ | ODR$_{BD-B} - \triangle$ODR$_{Sat-B}$ | 58.00 m/day |

event (14-15 July 2000) to the monthly mean values are 34.69% and 50.13% (36.45% and 68.95%) for Sat-A (Sat-B). The results of our simulation confirm the dependencies of the satellite drag on the ballistic coefficient and on the level of solar-geomagnetic activity. While such dependencies are generally intuitive our model is useful in quantifying these relationships and






can thus contribute to an improved situational awareness as well as mitigating the potential threat posed by solar-geomagnetic activity in modulating satellite trajectories.



*Competing interests.* The authors declare that they have no conflict of interest.

*Acknowledgements.* We acknowledge use of NASA/GSFC's Space Physics Data Facility's OMNIWeb service for the 1-hour averaged values
of $V_{sw}$, PD, Dst, IMF $B_y$ and $B_z$, and AE. The yearly-averaged SSN (version 1.0) was obtained from the World Data Center for the Sunspot
Index and Long-term Solar Observations (SILSO), Royal Observatory of Belgium, Brussels. The daily and monthly-averaged values for
SSN, F10.7 and Ap were provided by the GFZ German Research Centre for Geosciences and Space Weather Canada. Other data sources
include the Observatori de'Ebre (Spain) and the SOHO LASCO CME Catalog.





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
