# Peer review of "Atmospheric drag effects on modelled LEO satellites during the July 2000 Bastille Day event in contrast to an interval of geomagnetically quiet conditions"

_Annales Geophysicae, 2020_

## Referee Comment (RC1) · Anonymous Referee #1 · 4 Sep 2020

The paper is dealing with the impact of atmospheric drag on LEO satellites, which is a complex problem. This work does not contain any new ideas and the authors already acknowledge this fact in their abstract! Their methodology and some applications have been already presented in previous work. However, some of the data presented in the paper can be considered as new since the authors examine some specific cases. Overall, the paper does not provide significant scientific contribution, especially when compared with similar work from the literature.

[Figure]

Please also note the supplement to this comment:
https://angeo.copernicus.org/preprints/angeo-2020-33/angeo-2020-33-RC1-supplement.pdf

**Supplement:**

**Review of angeo-2020-33**

**General comments**

The paper is dealing with the impact of atmospheric drag on LEO satellites, which is a complex problem. This work does not contain any new ideas and the authors already acknowledge this fact in their abstract! Their methodology and some applications have been already presented in previous work. However, some of the data presented in the paper can be considered as new since the authors examine some specific cases. Overall, the paper does not provide significant scientific contribution, especially when compared with similar work from the literature.

The presentation is mostly clear though some part of the work could be elucidated further:

- It is generally accepted that it is quite challenging to accurate model drag effects on satellites and drag models include many parameters (e.g. drag coefficients, atmospheric density etc.) that are difficult to estimate accurately. The paper makes some assumptions w.r.t those parameters that are not always explained or justified properly (see specific points later).
- No information is provided about the methodology used for the numerical integration of the equations of motion with perturbation due to drag (i.e. coupled equations (1)-(4)).

The authors reach some conclusions that are not really new or substantial. The paper suggests that a model of satellite drag when combined with a high-fidelity atmospheric specification can lead to improved satellite ephemeris estimates. This is true but propagating orbits with high accuracy is complicated and one will also need to consider the aspherical potential of Earth's gravitational field (which is omitted here).

As a final point, it should be noted that all calculations in the paper have been done under the assumption that the satellite orbit is circular. However, real circular orbits of artificial satellites are possible only in the equatorial plane. It has been shown (see E. F. Jochim and M. C. Eckstein, "On the true circular orbit of a satellite, Celestial Mechanics", vol. 21, 149-153, 1980) that an inclined true satellite orbit cannot be circular because the satellite motion is influenced by a perturbing force as a result of the oblateness of the Earth (e.g. J2-term).

**Specific comments**

**Line 26:** "Atmospheric drag is the largest force affecting the motion of satellites in Low Earth Orbit (LEO) especially at altitudes below 800 km (Nwankwo et al., 2015)."

It is true that atmospheric drag most strongly influences the motion of a LEO satellite at low altitudes. However, perturbations due to the non-spherical shape of Earth are also important for LEOs and are in general considered for the orbit specification.

**Line 41:**
When presenting the formula of the drag force it is misleading to refer to the satellite speed (scalar quantity) instead of the velocity. The basic equation for aerodynamic drag shows that the associated force depends on the velocity of the satellite relative to the atmosphere. A simple assumption is to consider a mean motion due to the Earth's rotation. A more general expression could include wind variations (see

e.g. D. A. Vallado, "Fundamentals of Astrodynamics and Applications", 2007). It is not clear what is assumed in the paper (do the authors consider a static i.e. not rotating atmosphere?).

**Line 42:** "The satellite speed, vs, is a consequence of the balance between an inward-directed (towards earth) gravitational force at the satellite altitude and the outward-directed orbital centripetal force."

This is a strange statement! In a simple two-body problem (i.e. without perturbations), the gravitational force is the only force acting on the satellite (seen as a point mass). In classical (Newtonian) mechanics, we can define the centripetal force for uniform circular motion (still gravity is the only force acting on the point mass).

**Line 121:**
The cross sectional area (projected area) might be known for operational satellites but in principle is not so easy to calculate. For high precision studies, the satellite's attitude determination is employed for its calculation (near re-entry it is extremely difficult to know the satellites altitude accurately).

**Line 173:** The derivation of coupled equations (1)-(4) is not clear even after looking into the references mentioned in the paper… For example, how do the authors derive Eq. (2) in Nwankwo and Chakrabarti, 2014? In principle, one should be able to derive those equations starting from the general equation of motion for a two-body problem with perturbations (where $\mathbf{a}_p$ corresponds to the perturbing acceleration):

$$\ddot{\mathbf{r}} = -\frac{GM}{r^3}\mathbf{r} + \mathbf{a}_p$$

Assuming that the perturbations are only due to atmospheric drag we can write

$$\mathbf{a}_p = \mathbf{a}_{drag} = -\frac{1}{2}\frac{C_D A}{m}\varrho V_{rel}^2 \frac{\mathbf{V_{rel}}}{|\mathbf{V_{rel}}|}$$

However, as I have already mentioned the assumptions for the calculations of the relative velocity are not clear. Perhaps the authors could include an appendix in the paper with the derivation of the main equations used for calculating the satellite trajectory. The authors could also discuss briefly the methodology used for solving their set of coupled differential equations in order to obtain the instantaneous positions and velocities of the satellites.

**Line 186:** "…the radial distance, r, is used to model changes in satellite altitude."

The authors need to clarify their definition of satellite altitude. Does it take into consideration the oblate Earth or is it for example the radius of the orbit at a specific time minus the mean equatorial radius of the Earth?

Finally, NRLMSISE-00 takes as input a location normally provided in terms of latitude, longitude and height (altitude). It is not clear how this location is calculated from the satellite trajectory. The atmospheric density can change rapidly along the trajectory. What is the time step used for the estimation of the atmospheric drag parameters? Finally, the definition of the altitude (see also previous comment) could have an impact on the value of the atmospheric density and therefore on the calculated satellite drag.

**Technical corrections**

**Line 30:** Replace "(Nwankwo et al. (2015); and references therein)" with "(Nwankwo et al., 2015 and references therein)"

**Figure 1:** It should be clarified whether it shows the results of a simulation or it is just a cartoon demonstrating the effects of atmospheric drag on a satellite. If the former, then the parameters used for the simulation should be mentioned and possibly moved to section 3 (could also skip the pictures of the spacecraft).

**Line 81:** Replace "SunSpot Number" with "Sunspot Number" or "sunspot number" or "Sun Spot Number". You can always use the initials SSN but make sure you are consistent throughout the text (e.g. the caption of Figure 2 mentions Sun Spot Number…)

**Line 85:** "Figure 2 (after illustrates this cyclic variation in the monthly-averaged SSN along with the related solar-geophysical indices for the solar radio flux (F10.7) and the geomagnetic Ap." I assume that "(after" is a typo and needs to be removed.

**Line 131:** "Figure 3 is a plot of the 1-hour averaged variations in Vsw, PD, Dst, IMF By and Bz and AE for July 2006." You could consider including a reference for those data in the text and/or Figure 3 caption (e.g. in the Acknowledgment OMNIWeb service is mentioned)? Similar remark for Figure 4.

Consider replacing the word "UltraViolet" with "Ultraviolet" or "ultraviolet".

**Figure 10** seems to repeat what is show in figures **3** and **4**… If this is the case, then Figure 10 is redundant.

**Line 198:** "While a global specification was used to extract the density along the satellite flight path, the atmospheric curves used in Figures 5, 7 and 11 (to be discussed) to represent a general atmospheric response used a reference altitude of 450 km."

This sentence is a bit confusing. Consider revising.

**Table 2:** There is a typo in the units of Decay (it should be km instead of kg).

---

## Referee Comment (RC2) · Anonymous Referee #2 · 15 Sep 2020

The goal of the paper i.e The goal of this effort was to quantify how solar-geomagnetic activity influences atmospheric drag and perturbs satellite orbits, is very clear and worthwhile. The authors focused on the Bastille event because they have done similar work in another paper. I do not agree with the use of word as the authors seem to infer that they have not done any work different from the paper they first published on the topic. I reckon that the authors should have shown distinct comparison between the current paper and the previous paper and strongly justify why they focused on the bastille event.

---

## Author Comment (AC1) · 16 Sep 2020

We are particularly thankful to the reviewers for their time and effort. To begin our response to their highly esteemed comments we want to note (as they also noted) that the goal of this paper is very clear and without ambiguity. The title, abstract and conclusion also conveyed the specific accomplishments and what makes it differ from previous work. The statement that 'our findings are not particularly surprising' is not intended to make light of the accomplishment of this work. Neither does it imply or suggest that the 'work does not contain any new ideas'. Besides preceding the inclusion of a very im-

portant analysis, the statement is meant to 'whet the appetite' of the readers/scientific community preparatory to anticipated more profound findings that can lead to improved satellite ephemeris estimates using a new model that is now under formulation (which includes the results of analysis made in this paper).

We now highlight some important scientific contribution of this work (1) Given the great scientific interest in the Bastille Day great geomagnetic storm and its space weather consequences (particularly on orbital drag), this paper increases the visibility and better contribute to the scientific body of knowledge surrounding the Bastille Day events (2) this work also doubles as a strong review paper because it presented extensive details/review on atmospheric drag (and its relevance) in relation to solar activity, against properly referenced background of existing work. The significant number of readers who have interacted with this manuscript on this platform (and others) certainly did because of its relevance to them or their research. I am also aware of authors who have cited this paper in their new manuscript (3) the latter analysis (in this paper) that contrasted the interval of quiescent solar-geomagnetic activity with the Bastille Day event/perturbed condition is very instructive and important. This analysis motivated the development of new method and indices for description and estimation of drag effects on satellite ephemeris (comparing 2 regimes). We are now in the process of combining satellite drag model with high-fidelity atmospheric specification to produce such realistic estimation model (beginning with the results in this work).

Therefore, we do not agree that our paper does not provide significant scientific contribution. We would rather appreciate that the reviewer be generous to suggest content-wise inclusion or modification that will increase quality or form the ideas that the reviewer deems 'new' as an expert in lone with the purpose/goal of paper review. Thank you very much.

---

## Author Comment (AC2) · 16 Sep 2020

Your time and effort is highly esteemed and appreciated. Thank you for your insightful comment. We clarify that the statement that 'our findings are not particularly surprising' is not intended to make light of the accomplishment of this work. Neither does it imply or suggest that the work does not contain any new ideas. Besides preceding the inclusion of a very important analysis, the statement is meant to 'whet the appetite' of the readers/scientific community preparatory to anticipated more profound findings that can lead to improved satellite ephemeris estimates using a new model that is now

under formulation (which includes the results of analysis made in this paper). By highlighting some important scientific contribution of this work (below) I believe that we have been able to show 'distinct comparison between the current paper and the previous paper and strongly justify why we focused on the Bastille event.' The title, abstract and conclusion are replete and also convey the specific accomplishments and what makes it differ from previous work. Thank you very much.

Some important scientific contribution of this work

(1) Given the great scientific interest in the Bastille Day great geomagnetic storm and its space weather consequences (particularly on orbital drag), this paper will increase the visibility and better contribute to the scientific body of knowledge surrounding the Bastille Day events.

(2) this work also doubled as a strong review paper because it presented extensive details/review on atmospheric drag (and its relevance) in relation to solar activity, against properly referenced background of existing work. The significant number of readers who have interacted with this manuscript on this platform (and others) certainly did because of its relevance to them. I am also aware of authors have cited this paper in their new manuscript.

(3) the latter analysis (in this paper) that contrasted the interval of quiescent solar-geomagnetic activity with the Bastille Day event/perturbed condition is very instructive and important too. This analysis motivated the development of new method and indices for description and estimation of drag effects on satellite ephemeris (comparing 2 regimes). We are now in the process of combining satellite drag model high-fidelity atmospheric specification to produce such realistic estimation model (beginning with the results of this work).

---

## Referee Comment (RC3) · Anonymous Referee #2 · 17 Sep 2020

I am satisfied with the response from the author. I reckon that the authors could include the response, i.e " this work also doubled as a strong review paper because it presented extensive details/review on atmospheric drag (and its relevance) in relation to solar activity, against properly referenced background of existing work. The significant number of readers who have interacted with this manuscript on this platform (and others) certainly did because of its relevance to them. I am also aware of authors have cited this paper in their new manuscript" on the introduction. This will guide readers and clearly underpin the objectives of the present communication. It will be good for

the authors to spotlight the comparison of the Bastille day event to existing result right from the abstract. I advise that the part of the abstract that sort of infer that there are no new results in the paper should be taken off as it is grossly misleading if the authors claim that "this analysis motivated the development of new method and indices for description and estimation of drag effects on satellite ephemeris

How do you justify the below within in he current paper. You are supposed to convincingly show the strength in this paper for reviewing purposes. " We are now in the process of combining satellite drag model high-fidelity atmospheric specification to produce such realistic estimation model (beginning with the results of this work".

---

## Author Comment (AC3) · 25 Sep 2020

Thank you for the insightful comments/suggestions that are helping to improve the quality of this manuscript/work. We agree with the suggested content-wise inclusion, to better guide readers and clearly justify the objectives of the paper. Therefore, a better worded version of the suggested inclusion (that perfectly fits into the framework of the paper) will be added in the introduction (before the 'data and methodology' section) in the revised manuscript, Unless the editor suggest/advise otherwise. We also feel that some readers may misunderstand the phrase '...none of these findings were

particularly surprising or profound..' (as you noted). To avoid misinterpretation the sentence will be reworded (or removed and/or replaced) accordingly as suggested, to better convey the accomplishment of the work in the abstract.

Our analysis contrasted not only between the solar active and the quiescent regimes but also between the active regime and the Bastille day event, as well as between the quiescent regime and the Bastille day event. This kind of analysis enabled the definition of new indices presented in Table 3 and 4, from which h variations and ODRs were calculated/estimated for the model satellites. Because these calculations are based on model (which compared well with decay profile of some real satellites, as we reported in previous article e.g., Nwankwo et al. 2015) the next stage will be to obtain actual ephemeris data from satellite operators, for estimate model testing and comparison. Therefore, we shall report how we intend to progress from this point (as already explained) in the concluding part of this work. This way, we hope that we would have been able to justify the statement you noted within the current paper, and by extension further reflect the strength of this paper.

Thank you very much

---

## Author Comment (AC4) · 25 Sep 2020

While we so much appreciate your time and effort (which I take as a honour to our work), we continue to objectively analyse your comment in the light of the accomplishment and effort put into this work by the authors (most of whom are accomplished authors and technocrats). We feel that the comment "However, some of the data presented in the paper can be considered as new since the authors examine some specific cases" contradicts the claim that 'work does not contain any new ideas.' And if 'some of the data presented in the paper can be considered as new' it will be in the interest

of the scientific community (including several readers that have already interacted with this work) to direct effort towards helping to make this paper the best it can be (through this review process), instead of dwelling on the shortcoming (which of course can be improved). Thank you very much.

———————————————

---

## Author Response (AR2)

Journal: Annales Geophysicae
Title: Atmospheric drag effects on modelled LEO satellites during the July 2000 Bastille Day event in contrast to an interval of geomagnetically quiet conditions
Author(s): Victor U. J. Nwankwo et al.
Manuscript Number: angeo-2020-33
Manuscript Type: Regular paper
Iteration: Revision

17-10-2020

Dear Sir,

Thank you so much for your time and for organising the effort that is taking this manuscript through the stages making it a better and quality paper. We are also thankful to the reviews for their time and helpful comments and suggestions.

We have now revised our manuscript based on your suggestions, the reviewer comments and the answers we provided. While the revision lasted, we aimed at improving the paper's quality as much as possible in order to meet the standard of your reputed Journal. The point-by-point reply to the Editor and Reviewer's comments are detailed below:

**EDITOR'S COMMENTS**

Based on the reviewer comments and your answers, I invite you to submit a revised version of your manuscript. Please do take into account the detailed reviewer comments while preparing your revision. In particular, make sure that it is clear which contributions of your paper are new.

**Authors Response**

We have now clarified the contributions in this paper that are new as much as possible. To better track the clarifications, the reviewer comments and required adjustments are also provided (below). However, we note specific portions of the manuscript, which pin-points such contributions

1) In the Abstract: Besides the former inclusion (e.g., line 16-19), we buttressed and/or added to the contributions in line 19-21
2) In Section 1.2: In addition to the important detail provided under "Relevance of the study and its application" (e.g., line 116-119) we further provided more, as well as the justification for focusing on the Bastille Day event in line 120-133
3) In Section 4: line 316-322.

**COMMENTS BY REFEREE #1**

The paper is dealing with the impact of atmospheric drag on LEO satellites, which is a complex problem. This work does not contain any new ideas and the authors already acknowledge this fact in their abstract! Their methodology and some applications have been already presented in previous work. However, some of the data presented in the paper can be considered as new since the authors examine some specific cases. Overall, the paper does not provide significant scientific contribution, especially when compared with similar work from the literature.

**Authors Response**

Although the reviewer admitted that some of the data we presented in the paper are new, he stopped short of acknowledging the accomplishment of this effort. The referee feels that our paper does not provide

significant scientific contribution. We respect the judgment of the referee and do not take it a being biased. However, we beg to differ on the claim. As the goal of manuscript review (by Referees) are generally to 'provide unbiased and constructive comments aimed, whenever possible, at improving the work,' we feel that a generous suggestion of content-wise inclusion or modification that will increase quality would have been a better conclusion of the referee. To address the comment of the referee, I think elucidating the new scientific contributions of this paper in clear terms (as also suggested by the Editor) is the best thing to do. We therefore highlight them below (as was done earlier via our response to RC1).

(1) This paper emphasized and/or focused on the Bastille Day great geomagnetic storm (and associated phenomena). It is hoped that efforts directed towards assessing, monitoring, modeling and/or prediction of the impacts associated with sudden severe solar energetic transients (like this one) are key to mitigating the potential threat posed by such event in future occurrence. This is the first time we are modeling atmospheric drag effect associated with the Bastille Day event (BDE). Therefore, this paper increases the visibility and better contribute to the scientific body of knowledge surrounding the BDE (as earlier stated). Please see the specific portions above where we provided the detail of this contribution (and others) in the text, as well as the justification for focusing on the BDE.

(2) In our analysis we used new method and indices to describe and estimate drag effects on the satellite trajectory when contrasting between the (i) solar active and the quiescent regimes (ii) active regime and the Bastille day storm, and (iii) the quiescent regime and the Bastille day event/storm. This analysis and the results obtained is now helping us to produce estimation model that compares effects between regimes of varying solar-geomagnetic activity. In addition to examining a specific case (different from previous study), we used a relatively new (or novel) approach/method. As much as I prefer to keep a low profile on this at this stage, I am yet to find similar approach in literature till date.

(3) This work doubles as a strong review paper. We presented extensive details/review on atmospheric drag (and its relevance) in relation to solar activity, against properly referenced background of existing work. If carefully perused, one could see a concise comprehensive connection between atmospheric drag and solar-geomagnetic activity that is particularly unprecedented when compared with our previous work (not overall literature in the area) – thanks to the proficiency of some co-authors!

In summary, our pattern of result presentation (especially in abstract) may look similar but we believe the accomplishments or contributions of this effort are replete and should not be overemphasized (especially when one carefully read beyond the abstract).

**COMMENTS BY REFEREE #2**

**1.** The goal of the paper i.e The goal of this effort was to quantify how solar-geomagnetic activity influences atmospheric drag and perturbs satellite orbits, is very clear and worthwhile. The authors focused on the Bastille event because they have done similar work in another paper. I do not agree with the use of word as the authors seem to infer that they have not done any work different from the paper they first published on the topic. I reckon that the authors should have shown distinct comparison between the current paper and the previous paper and strongly justify why they focused on the bastille event.

**Authors Response**

We have now revised the manuscript in the manner that buttressed the scientific contribution in this work that are new and also different from previous work (please also see the above highlight). This way the 'distinct comparison between the current paper and the previous paper' can be clearly understood. We have also justified the reason we focused on the Bastille event. The specific portions of the manuscript which pin-points such inclusions are listed below

1) In the Abstract: line 19-21 (in addition to line 16-19)
2) In Section 1.2: line 120-133 (In addition to line 116-119)
3) In Section 4: line 316-322.

**2.** I am satisfied with the response from the author. I reckon that the authors could include the response, i.e "this work also doubled as a strong review paper because it presented extensive details/review on atmospheric drag (and its relevance) in relation to solar activity, against properly referenced background of existing work. The significant number of readers who have interacted with this manuscript on this platform (and others) certainly did because of its relevance to them. I am also aware of authors have cited this paper in their new manuscript" on the introduction. This will guide readers and clearly underpin the objectives of the present communication.

***Response*** Please see line 131-133 for the inclusion. We excluded a few lines since this is a scientific article

It will be good for the authors to spotlight the comparison of the Bastille day event to existing result right from the abstract.

***Response*** Please see line 3-4, 18, 19-22

I advise that the part of the abstract that sort of infer that there are no new results in the paper should be taken off as it is grossly misleading if the authors claim that "this analysis motivated the development of new method and indices for description and estimation of drag effects on satellite ephemeris

***Response*** The suggested phrase have now been removed from the abstract.

How do you justify the below within in he current paper. You are supposed to convincingly show the strength in this paper for reviewing purposes. "We are now in the process of combining satellite drag model high-fidelity atmospheric specification to produce such realistic estimation model (beginning with the results of this work".

***Response*** In addition to examining a specific case (the BDE) that is different from previous study (with new results), one other strength of this work is that we used a relatively new approach that is now helping us to produce estimation model that compares effects between regimes of varying solar-geomagnetic activity – and such formulation is a process!

In conclusion, we have also closely perused the manuscript again to eliminate or reduce typographical errors and expressions that can make the understanding of any portion difficult for the readers (as much as possible). We believe that in its current state, our revised manuscript is now suitable for further consideration by your journal, and sincerely hope that the paper will now be accepted for publication.

Thank you very much.

Victor U. J. Nwankwo

[revised manuscript text omitted]

---

## Author Response (AR3)

Journal:  Annales Geophysicae
Title: Atmospheric drag effects on modelled LEO satellites during the July 2000 Bastille Day event in contrast to an interval of geomagnetically quiet conditions
Author(s): Victor U. J. Nwankwo et al.
Manuscript Number: angeo-2020-33
Manuscript Type: Regular paper
Iteration: Revision

28-01-2021

Dear Editor,

Thank you so much for handling our manuscript and for the effort that is helping us to improve on the quality of this work. We are also thankful to the reviews for their time and helpful comments and suggestions. The point-by-point reply to the Editor and Reviewer's comments are detailed below:

**EDITOR'S COMMENTS**

Dear authors,
Apparently you seem to have missed the detailed reviewer remarks provided by one of the reviewers during the previous iteration. I therefore return the manuscript back to you with the request to address these; see the interactive discussion RC1, see
https://angeo.copernicus.org/preprints/angeo-2020-33/angeo-2020-33-RC1-supplement.pdf
Once I receive your response and the updated manuscript, the editorial process can continue.

*Authors Response*

I admit we did not notice this supplementary comments/report attached to RC1 by the reviewer #1 until you drew our attention to it in your last communication. We sincerely apologise for this oversight. While our response during the interactive discussion were based on short comments only, we have now revised our manuscript accordingly.

**COMMENTS BY REFEREE #1**
General comments
The paper is dealing with the impact of atmospheric drag on LEO satellites, which is a complex problem. This work does not contain any new ideas and the authors already acknowledge this fact in their abstract! Their methodology and some applications have been already presented in previous work. However, some of the data presented in the paper can be considered as new since the authors examine some specific cases. Overall, the paper does not provide significant scientific contribution, especially when compared with similar work from the literature.

*Authors Response*

We appreciate and applaud the effort of the referee, working through this manuscript. However, we beg to differ on the claim that this work 'does not provide significant scientific contribution.'As the lead-author my experience may be far below the expertise of the referee, but I believe our work is both significant and evolving, and a process (of review) like this should help to advance the quality of the work. We highlight the scientific contributions of this paper in clear terms as follows.

(1) This paper emphasized and/or focused on the Bastille Day great geomagnetic storm (and associated phenomena). It is hoped that efforts directed towards assessing, monitoring, modeling and/or prediction of the impacts associated with sudden severe solar energetic transients (like this one) are key to mitigating

the potential threat posed by such event in future occurrence. This is the first time we are modeling atmospheric drag effect associated with the Bastille Day event (BDE). Therefore, this paper increases the visibility and better contribute to the scientific body of knowledge surrounding the BDE (as earlier stated).

(2) In our analysis we used new method and indices to describe and estimate drag effects on the satellite trajectory when contrasting between the (i) solar active and the quiescent regimes (ii) active regime and the Bastille day storm, and (iii) the quiescent regime and the Bastille day event/storm. This analysis and the results obtained is now helping us to produce estimation model that compares effects between regimes of varying solar-geomagnetic activity. In addition to examining a specific case (different from previous study), we used a relatively new approach/method. As much as I prefer to keep a low profile on this at this stage, I am yet to find similar approach in literature.

(3) This work doubles as a review paper. We presented extensive details/review on atmospheric drag (and its relevance) in relation to solar activity, against properly referenced background of existing work. If carefully perused, one could see a concise and comprehensive connection between atmospheric drag and solar-geomagnetic activity that is particularly unprecedented when compared with our previous work (not overall literature in the area).

**Comments**
The presentation is mostly clear though some part of the work could be elucidated further:

- It is generally accepted that it is quite challenging to accurate model drag effects on satellites and drag models include many parameters (e.g. drag coefficients, atmospheric density etc.) that are difficult to estimate accurately. The paper makes some assumptions w.r.t those parameters that are not always explained or justified properly (see specific points later).
- No information is provided about the methodology used for the numerical integration of the equations of motion with perturbation due to drag (i.e. coupled equations (1)-(4)).

*Authors response*: It is true that some parameters are difficult to accurately estimate when modeling drag effects. We acknowledge and appreciate the expertise of the reviewer and prefer to see these constructive comments ultimately aiming at improving not only this manuscript but also our research effort in this direction. However, we also know that the best of models has both approximations and assumptions, and despite such and many 'difficulties' associated with accurate estimation, models have not only evolved but also contributed to the overwhelming breakthroughs made so far in space research and technology. Our work has evolved (while building on our previous work and many other insightful work by other authors), just as the best of works also evolved. Our previous work acknowledged the challenges associated with accurate estimation or determination of parameters associated with modeling drag effects on satellites. For examples, we discussed the difficulties associated with estimating drag coefficient in page 49 of Nwankwo and Chakrabarti (2014) and atmospheric density on the same article (and page), page 48 of Nwankwo et al. (2015), and chapter 49, pp 6 of Nwankwo (2018). A bigger problem confronting the estimation of atmospheric density is the fact that the individual effects of various solar forcing mechanisms that causes fluctuations in neutral and ionized density are even more difficult to estimate and/or model (Kutiev et al., 2013; Nwankwo et al. 2015). Yet, models continue to lead the frontiers in space exploration and exploitation, despite the large body of work highlighting these limitations. Your comments/report is important to us – we will continue to build and advance our work accordingly, taking the needful into consideration.

On the methodology used for the numerical integration of the equations of motion, we explained in Nwankwo and Chakrabarti (2014) that the sets of differential equations were solved by fourth order Runge-Kutta method. Given the need for a concise manuscript, we thought a repetition of the procedures

would make it lengthy. However, we have once more discussed in brief the methodology used for solving their set of coupled differential equations. Please see Line 223-228.

**Comments**

The authors reach some conclusions that are not really new or substantial. The paper suggests that a model of satellite drag when combined with a high-fidelity atmospheric specification can lead to improved satellite ephemeris estimates. This is true but propagating orbits with high accuracy is complicated and one will also need to consider the aspherical potential of Earth's gravitational field (which is omitted here).

*Authors response*: In this work, the density profile was derived from Naval Research Laboratory Mass Spectrometry and Incoherent Scatter Extended 2000 (NRLMSISE-00) empirical atmospheric model, and this 'model accounted for the approximate spheroidal symmetry of the Earth and the atmosphere by incorporating a gravity field and an effective Earth radius which are both latitude-dependent and by using spherical harmonics to represent spatial variability of the key parameters that define temperature and species number density profiles.' Please see Picone et al 2002. Also, see section 3 (from line 194-210). We have now noted other perturbing forces (including the aspherical potential of Earth's gravitational field) and clearly stated the scope of our paper for clarity.

**Comments**

As a final point, it should be noted that all calculations in the paper have been done under the assumption that the satellite orbit is circular. However, real circular orbits of artificial satellites are possible only in the equatorial plane. It has been shown (see E. F. Jochim and M. C. Eckstein, "On the true circular orbit of a satellite, Celestial Mechanics", vol. 21, 149-153, 1980) that an inclined true satellite orbit cannot be circular because the satellite motion is influenced by a perturbing force as a result of the oblateness of the Earth (e.g. J2-term).

*Authors response*: We have continue to build our model to incorporate many complex orbit parameters, towards a robust model with good approximations. As we work towards extending our work to other orbit types, we feel the consideration of circular (or near-circular) orbit in this work should be treated as a matter of choice rather than 'a weak point.' The fact that 'real circular orbits for artificial satellites are possible' makes this choice relevant and applicable. Among others, CHAMP was launched into circular, near-polar orbit (Reigber et al. 2002) and GOCE was launched into Sun-synchronous, circular, dawn-dusk low Earth orbit (Johannessen et al., 2003). There are also numerous mention and applications of circular orbit satellites in standard texts/work (e.g., Sidi, 1997; Larson and Wertz, 1999; Leonard et al. 2012). Nonetheless, our future work will prioritize analysis of perturbations in other types of orbit.

**Comments**

Specific comments
Line 26: "Atmospheric drag is the largest force affecting the motion of satellites in Low Earth Orbit (LEO) especially at altitudes below 800 km (Nwankwo et al., 2015)."

It is true that atmospheric drag most strongly influences the motion of a LEO satellite at low altitudes. However, perturbations due to the non-spherical shape of Earth are also important for LEOs and are in general considered for the orbit specification.

*Authors response:* We have now noted/included the importance of such perturbations (due to non-spherical shape of Earth) in the text. Please see inclusion in line 30-32 and 194-210.

**Comments**

Line 41:
When presenting the formula of the drag force it is misleading to refer to the satellite speed (scalar quantity) instead of the velocity. The basic equation for aerodynamic drag shows that the associated force depends on the velocity of the satellite relative to the atmosphere. A simple assumption is to consider a mean motion due to the Earth's rotation. A more general expression could include wind variations (seee.g. D. A. Vallado, "Fundamentals of Astrodynamics and Applications", 2007). It is not clear what is assumed in the paper (do the authors consider a static i.e. not rotating atmosphere?).

Line 42: "The satellite speed, vs, is a consequence of the balance between an inward-directed (towards earth) gravitational force at the satellite altitude and the outward-directed orbital centripetal force."

This is a strange statement! In a simple two-body problem (i.e. without perturbations), the gravitational force is the only force acting on the satellite (seen as a point mass). In classical (Newtonian) mechanics, we can define the centripetal force for uniform circular motion (still gravity is the only force acting on the point mass).

*Authors response:* We have now replaced the satellite's 'speed' with 'velocity' accordingly. The sentence of line 42 is actually true and never out of context. We really do not know how best to express this to the satisfaction of the referee... Perhaps it may be preferable to say that Earth's gravitational force provides the centripetal force of the satellite. $F_G = F_C$ or $GMm/R^2 = mv^2/R$ (from which the velocity, $v_s$, of the satellite is derived). However, we have now removed that portion to avoid any confusion.

**Comments**
Line 121:
The cross sectional area (projected area) might be known for operational satellites but in principle is not so easy to calculate. For high precision studies, the satellite's attitude determination is employed for its calculation (near re-entry it is extremely difficult to know the satellites altitude accurately).

*Authors' response*: Thanks for this important point. We have been added it to text. Please see inclusion in line 38-39.

**Comments**
**Line 173:** The derivation of coupled equations (1)-(4) is not clear even after looking into the references mentioned in the paper… For example, how do the authors derive Eq. (2) in Nwankwo and Chakrabarti, 2014? In principle, one should be able to derive those equations starting from the general equation of motion for a two-body problem with perturbations (where $\mathbf{a}_p$ corresponds to the perturbing acceleration):

$$\ddot{\mathbf{r}} = -\frac{GM}{r^3}\mathbf{r} + \mathbf{a}_p$$

Assuming that the perturbations are only due to atmospheric drag we can write

$$\mathbf{a}_p = \mathbf{a}_{drag} = -\frac{1}{2}\frac{C_D A}{m}\varrho V_{rel}^2 \frac{\mathbf{V}_{rel}}{|\mathbf{V}_{rel}|}$$

However, as I have already mentioned the assumptions for the calculations of the relative velocity are not clear. Perhaps the authors could include an appendix in the paper with the derivation of the main equations used for calculating the satellite trajectory. The authors could also discuss briefly the methodology used for solving their set of coupled differential equations in order to obtain the instantaneous positions and velocities of the satellites.

**Authors Response: We believe that a scientific research article should be precise and concise with a well defined objective/goal. We have clearly defined the scope of the problem this work intends to solve. While the points raised here are important, we are careful not to take this work out of context (or beyond scope) and make it voluminous. Be it third-body perturbations, the oblateness of the Earth, orbit propagation, gravitational perturbations etc., related work dealt with specific and clearly defined problem (e.g., Domingos et al., 2008; Leonard et al., 2012; Shou 2014; Sanjeeviraja et al., 2018). Similarly, this paper concentrates on atmospheric drag effects satellites. However, we are working on a book that incorporates aspects of our work covered in these areas and those not yet cover (including aspects generously suggested by referees).**

**The results of this coupled equations (in spherical coordinate system) used in our work compared very well with other formulations (or equation of motion used in literature) and certainly derivable from the general equation of motion for a two-body problem with perturbations. We have discussed in brief the methodology used for solving the set of coupled differential equations. Please see Line 223-228. Some authors have made and included derivation in related work (e,g., Shou 2014). However, we crave the understanding of the referee/editor to allow us make such inclusion in future work (already determined and in progress).**

**Comments**
Line 186: "...the radial distance, r, is used to model changes in satellite altitude."
The authors need to clarify their definition of satellite altitude. Does it take into consideration the oblate Earth or is it for example the radius of the orbit at a specific time minus the mean equatorial radius of the Earth?

Finally, NRLMSISE-00 takes as input a location normally provided in terms of latitude, longitude and height (altitude). It is not clear how this location is calculated from the satellite trajectory. The atmospheric density can change rapidly along the trajectory. What is the time step used for the estimation of the atmospheric drag parameters? Finally, the definition of the altitude (see also previous comment) could have an impact on the value of the atmospheric density and therefore on the calculated satellite drag.

*Authors Response*: **We have clearly defined r in line 220 of the manuscript, just as we clarified the scope and/or limitation of the study in line 201-206. The NRLMSISE-00 empirical atmospheric model incorporated into our drag model such that the instantaneous motion and position of the satellites varies in consonance with the density of the location through which the satellites traverses. Please see the curves in our results; notice the resultant perturbations in the satellites height and orbit decay rate (ODR)) due to temperature and density fluctuations. In Nwankwo and Chakrabarti, 2018 we also demonstrated how satellites trajectory are affected by latitudinal changes as a result of density variations.**

**Technical corrections**

**Line 30**: Replace "(Nwankwo et al. (2015); and references therein)" with "(Nwankwo et al., 2015 and references therein)"

*Authors response*: "(Nwankwo et al. (2015); and references therein)" have been replaced with "(Nwankwo et al., 2015 and references therein)". Please see line 34-35

**Figure 1**: It should be clarified whether it shows the results of a simulation or it is just a cartoon

demonstrating the effects of atmospheric drag on a satellite. If the former, then the parameters used for the simulation should be mentioned and possibly moved to section 3 (could also skip the pictures of the spacecraft).

*Authors' response*: We have now clarified that that Figure 1 is actually a result of our simulation, and the figure has been moved to section 3. Please see lne 223-228

**Line 81**: Replace "SunSpot Number" with "Sunspot Number" or "sunspot number" or "Sun Spot Number". You can always use the initials SSN but make sure you are consistent throughout the text (e.g. the caption of Figure 2 mentions Sun Spot Number...)

*Authors' response*: "SunSpot Number" has now been replaced with "Sunspot Number", please see line 81 and new Figure 1 caption.

**Line 85**: "Figure 2 (after illustrates this cyclic variation in the monthly-averaged SSN along with the related solar-geophysical indices for the solar radio flux (F10.7) and the geomagnetic Ap." I assume that "(after" is a typo and needs to be removed.

*Authors' response*: The typo error has been corrected accordingly. Please see line 85-86

**Line 131**: "Figure 3 is a plot of the 1-hour averaged variations in Vsw, PD, Dst, IMF By and Bz and AE for July 2006." You could consider including a reference for those data in the text and/or Figure 3 caption (e.g. in the Acknowledgment OMNIWeb service is mentioned)? Similar remark for Figure 4.

*Authors' response*: We have now included a reference for the data used in the text/figures. Please see new Figures 2 and 3 captions.

Consider replacing the word "UltraViolet" with "Ultraviolet" or "ultraviolet".

*Authors' response*: The word "UltraViolet" have been replaced with "Ultraviolet".

**Figure 10** seems to repeat what is show in figures 3 and 4... If this is the case, then Figure 10 is redundant.

*Authors response*: Figure 10 is not exactly same as figures 3 and 4 (now 2 and 3). Please see the time axes.

**Line 198**: "While a global specification was used to extract the density along the satellite flight path, the atmospheric curves used in Figures 5, 7 and 11 (to be discussed) to represent a general atmospheric response used a reference altitude of 450 km."

This sentence is a bit confusing. Consider revising.
*Authors' response*: The sentence has been removed to avaoid any form of confusion.

Table 2: There is a typo in the units of Decay (it should be km instead of kg).

*Authors' response*: The typo has been corrected accordingly.

**COMMENTS BY REFEREE #2**

**1.** The goal of the paper i.e The goal of this effort was to quantify how solar-geomagnetic activity influences atmospheric drag and perturbs satellite orbits, is very clear and worthwhile. The authors focused on the Bastille event because they have done similar work in another paper. I do not agree with the use of word as the authors seem to infer that they have not done any work different from the paper they first published on the topic. I reckon that the authors should have shown distinct comparison between the current paper and the previous paper and strongly justify why they focused on the bastille event.

*Authors Response*

We have now revised the manuscript in the manner that buttressed the scientific contribution in this work that are new and also different from previous work (please also see the above highlight). This way the 'distinct comparison between the current paper and the previous paper' can be clearly understood. We have also justified the reason we focused on the Bastille event. The specific portions of the manuscript which pin-points such inclusions are listed below

1) In the Abstract: line 19-21 (in addition to line 16-19)
2) In Section 1.2: line 118-133
3) In Section 4: line 342-344.

**2.** I am satisfied with the response from the author. I reckon that the authors could include the response, i.e "this work also doubled as a strong review paper because it presented extensive details/review on atmospheric drag (and its relevance) in relation to solar activity, against properly referenced background of existing work. The significant number of readers who have interacted with this manuscript on this platform (and others) certainly did because of its relevance to them. I am also aware of authors have cited this paper in their new manuscript" on the introduction. This will guide readers and clearly underpin the objectives of the present communication.

*Response* Please see line 131-133 for the inclusion. We excluded a few lines since this is a scientific article

It will be good for the authors to spotlight the comparison of the Bastille day event to existing result right from the abstract.

*Response* Please see line 3-4, 18, 19-22

I advise that the part of the abstract that sort of infer that there are no new results in the paper should be taken off as it is grossly misleading if the authors claim that "this analysis motivated the development of new method and indices for description and estimation of drag effects on satellite ephemeris

*Response* The suggested phrase have now been removed from the abstract.

How do you justify the below within in he current paper. You are supposed to convincingly show the strength in this paper for reviewing purposes. "We are now in the process of combining satellite drag model high-fidelity atmospheric specification to produce such realistic estimation model (beginning with the results of this work".

*Response* In addition to examining a specific case (the BDE) that is different from previous study (with new results), one other strength of this work is that we used a relatively new approach that is now helping us to produce estimation model that compares effects between regimes of varying solar-geomagnetic

activity – and such formulation is a process!

In conclusion, we have also closely perused the manuscript again to eliminate or reduce typographical errors and expressions that can make the understanding of any portion difficult for the readers (as much as possible). We believe that in its current state, our revised manuscript is now suitable for further consideration by your journal, and sincerely hope that the paper will now be accepted for publication.

Thank you very much.

Victor U. J. Nwankwo

**References**

Domingos R. C., R. Vilhena de Moraes, and A. F. Bertachini De Almeida Prado, 2008. Third-Body Perturbation in the Case of Elliptic Orbits for the Disturbing Body. Hindawi Publishing Corporation Mathematical Problems in Engineering Volume 2008, Article ID 763654, 14 pages doi:10.1155/2008/763654

Johannessen J.A., G. Balmino, C. Le Provost, R. Rummel, R. Sabadini, H. Sünkel, C. C. Tscherning, P. Visser, P. Woodworth, C. W. Hughes, P. Legrand, N. Sneeuw, F. Perosanz, M. Aguirre-Martinez, H. Rebhan And M. R. Drinkwater, 2003. The European Gravity Field And Steady-State Ocean Circulation Explorer Satellite Mission: Its Impact On Geophysics. Surveys in Geophysics 24: 339-386, 2003.

Kutiev, I., Tsagouri, Ioanna, Perrone, Loredana, Pancheva, Dora, Mukhtarov, Plamen, Mikhailov, Andrei, Lastovicka, Jan, Jakowski, Norbert, Buresova, Dalia, Blanch, Estefania, Andonov, Borislav, Altadill, David, Magdaleno, Sergio, Parisi, Mario, Torta, Joan Miquel, 2013. Solar activity impact on the Earth's upper atmosphere. J. Space Weather Space Clim. 3, A06.

Leonard, J.M., Forbes, J., Born, G.H., 2012. Impact of tidal density variability on orbital and reentry predictions. Space Weather 10, S12003.

Picone, J.M., Hedin, A.E., Drob, D.P., Aikin, A.C., 2002. NRLMSISE-00 empirical model of the atmosphere: Statistical comparisons and scientific issues. J. Geophys. Res. 107, 1468.

Reigber Ch., H. Ltihr and P. Schwintzer, 2002. Champ Mission Status. Adv. Space Res. 30, 129-134.

Shou H-N, 2014. Orbit Propagation and Determination of Low Earth Orbit Satellites. Hindawi Publishing Corporation International Journal of Antennas and Propagation Volume 2014, Article ID 903026, 12 pages http://dx.doi.org/10.1155/2014/903026

Sanjeeviraja T., Santhanakrishnan R. and Lakshmi S., 2018. Study the Influence of Atmospheric Drag and $J_2$ Effect in a Close-proximity Operation at LEO. International Journal of Engineering & Technology, 7 (4.36), 403-408.

---

## Author Response (AR4)

Journal:  Annales Geophysicae
Title: Atmospheric drag effects on modelled LEO satellites during the July 2000 Bastille Day event in contrast to an interval of geomagnetically quiet conditions
Author(s): Victor U. J. Nwankwo et al.
Manuscript Number: angeo-2020-33
Manuscript Type: Regular paper
Iteration: Correction

04-03-2021

Dear Editor,

We are deeply grateful to you (and team) for handling our manuscript, and most importantly for coordinating the efforts that helped us improve the quality of this work, which has now culminated to the acceptance of our manuscript for publication. We really cannot thank the reviews enough but their time and effort put into this review process are both applauded and appreciated. Their helpful comments and suggestions in no small measure led to the ultimate achievement of the goal of this review process - improving the work.

We are pleased to inform you that the indicated technical corrections has been accordingly. The list of required corrections and the respective responses are detailed below:

**EDITOR'S COMMENTS**

Comments to the Author:
Dear authors,

It is my pleasure to inform you that - based on the recommendations of the reviewers - we are now ready to publish your paper, provided that you make just a few technical corrections as indicated below:

• The symbol for solar wind speed in the text needs to be checked. It seems that something went wrong with the subscript and the letter W does not appear in the correct position.

**Response**: The symbol for solar wind ($V_{sw}$) has been checked and corrected accordingly. Please see lines 150 and 153

• In equation (1) the acceleration term is missing a bar above r. Personally, I prefer to use bold letters to indicate vectors e.g. as is done in Nwankwo PhD thesis (2016) equation 3-6. However, the authors are free to choose a style as long as they are consistent.

**Response**: Equation (1) has been corrected accordingly. Please see line 192

• I suggest to rephrase the sentence "Our model cartered for the effect of Earth's gravity (since the derivation of the satellites' velocity is based on the balance between the gravitational and centripetal force)." in line 199. I think I understand what the authors are trying to say here but the wording might cause some confusion (there is also a typo in the sentence) because the centripetal force is a gravitational force. To avoid any confusion, I suggest to write the following (or something similar) instead: "Our model takes into account the effect of Earth's gravity since the derivation of the satellites' velocity is based on the concept of the centripetal force."

**Response**: As suggested the sentence "Our model cartered for the effect of Earth's gravity (since the

derivation of the satellites' velocity is based on the balance between the gravitational and centripetal force)" has been rephrased  to read "Our model takes into account the effect of Earth's gravity since the derivation of the satellites' velocity is based on the concept of the centripetal force." Please see linea 199-200.

• In Figure 4, I would personally remove the image of Earth to emphasise the fact that the plot shows the results of a simulation (nevertheless, the authors should feel no obligation to change the figure).

**Response**: The image of the Earth has been removed from figure 4 accordingly.

• Earth appears mostly capitalised but not in lines 69, 182, 212, 326.

**Response**: We have now capitalised the work 'Earth' (and 'Sun') were necessary (as indicated). Please see lines 37, 61, 69, 181, 212 and 329.

We believe that in its current state, our corrected manuscript is now suitable for publication, and we look forward to having our it published at the earliest time possible.

Thank you very much.

Victor U. J. Nwankwo